# Boiler Combustion Optimization of Vegetal Crop Residues from Greenhouses

José Vicente Reinoso Moreno [1,*] , María Guadalupe Pinna Hernández [1,2] , María Dolores Fernández Fernández [3], Jorge Antonio Sánchez Molina [4] , Juan Carlos López Hernández [3] and Francisco Gabriel Acién Fernández [1]

1   Department of Chemical Engineering, University of Almería, Carretera de Sacramento, La Cañada de San Urbano, 04120 Almería, Spain; gpinnahernandez@ual.es (M.G.P.H.); facien@ual.es (F.G.A.F.)
2   Solar Energy Research Centre (CIESOL), Joint Centre University of Almería-CIEMAT, 04120 Almería, Spain
3   Las Palmerillas Experimental Station, Cajamar Caja Rural Foundation, 04710 Almería, Spain; mdoloresfernandez@fundacioncajamar.com (M.D.F.F.); jcpalmerillas@gmail.com (J.C.L.H.)
4   Automatic Control, Robotics and Mechatronic Research Group (TEP 197), Department of Informatics, University of Almería, Carretera de Sacramento, not numbered, 04120 Almería, Spain; jorgesanchez@ual.es
*   Correspondence: rmj519@ual.es; Tel.: +34-717708907

**Abstract:** This work presents an alternative for adding value to greenhouse crop residues, used for (1) heating and (2) as a $CO_2$ source. Both options are focused on greenhouse agricultural production, but could be applied to other applications. The influence of factors, such as the air/fuel rate and turbulence inside the combustion chamber, is studied. Our results show that for pine pellets, olive pits, tomato-crop residues, and a blend of the latter mixed with almond prunings (75–25%), the thermal losses ranged from 19.5–53.1, 20.5–58.9, 39.9–95%, and 29.4–75.5%, respectively, while the $NO_X$ emissions were 30–247, 411–1792, and 361–2333 mg/Nm$^3$, respectively. The above-mentioned blend was identified as the best set-up. The thermal losses were 39.2%, and the CO, $NO_X$, and $SO_2$ concentrations were 11,690, 906, and 1134 mg/Nm$^3$, respectively (the gas concentration values were recalculated for 0% $O_2$). Currently, no other work exists in the literature include a similar analysis performed using a boiler with a comparable thermal output (160.46 kW). The optimal configurations comply with the relevant local legislation. This optimization is important for future emission control strategies relating to using crop residues as a $CO_2$ source. The work also highlights the importance of ensuring a proper boiler set-up for each case considered.

**Keywords:** biomass combustion; boiler efficiency; waste valorization; $CO_2$ storage; heating applications; waste heat source

## 1. Introduction

There are several alternatives for adding value to crop-generated biomass [1–6]. One available alternative consists of using biomass as an energy source. However, this has not been fully developed. The European Union's objectives for the renewable energy consumption fraction for 2020 have almost been met, although this fraction has now been raised from 20 to 32% for 2030 (at the moment, the fractions are 18.8 and 17.5%, for the EU and Spain, respectively) [7]. Biomass offers an additional way to increase this fraction. Such an option has been contemplated in several national plans, Spain's being one of them [8]. Agricultural residues are one of the potential biomass sources out of the various candidates currently available. Nevertheless, its consumption rate could be increased.

Horticultural fruit growing is an important economic activity in Almeria Province (south-eastern Spain [9–11]). Of the crops grown in this zone, tomato (*Solanum Lycopersicum*) is the most widespread. Such extensive horticultural fruit production in a specific zone also leads to important amounts of vegetal residues. One portion of the residues generated corresponds to the aerial part of these plants. This waste is discarded after the plant's productive life is over. Differences in distribution from season to season are an additional

issue [12,13]. This article studies the application of this biomass source to greenhouse heating and $CO_2$ enrichment. The amount of energy that could be obtained annually in this zone is $1.16 \times 10^5$ TOE (from tomato and other crops). This value was estimated based on the amount generated [14,15], and its calorific value. Moreover, it was considered that the biomass moisture content would be reduced to 21.3% [15]. Currently, ENCE (a private company) is carrying out a project where they plan to utilize this biomass for electricity generation [16]. Applying it to combustion equipment in greenhouses, with its moderate thermal output, has yet to be developed. On the other hand, the utilization of other types of biomass is further advanced (i.e., pine pellets), although this biomass is generated in other zones, so utilizing it involves additional transportation costs and a higher $CO_2$ footprint. Regarding $CO_2$ enrichment, this alternative is already used in similar applications where photosynthetic organisms are grown, such as in aquaculture [17–19]. Using this biomass for $CO_2$ enrichment in greenhouses is also an interesting option [20].

Concerning the biomass combustion efficiency in boilers, this case can be considered as a process of energy transference. The flue gas generated from biomass combustion transfers thermal heat to the boiler's water volume. The temperature gradient must be high enough to maintain a certain energy transfer rate. Thus, there is an optimum flue-gas temperature value. When used for $CO_2$ enrichment, one must consider the other compounds generated when this biomass is combusted, including $SO_2$ and $NO_X$, as their presence can have a considerably negative impact on plant growth. This can manifest in different ways depending on the plant species. The most common effect is chlorosis and necrosis of the leaves. In other cases, it leads to an appreciable reduction in growth [20,21]. At the same time, CO levels can be used as an indicator of $O_2$ supply. The C/O ratio is also related to the production of organic compounds [22,23]. For these reasons, the optimization focused on these three compounds. Nevertheless, there are other relevant emissions to take into account, such as particles and organic compounds. These gaseous emissions are influenced by the combustion device settings, although other factors related to the design of these devices significantly affect performance, such as the combustion chamber size, the position, and orientation of the air inlets, the airflow pattern, the flue-gas residence time, the position and method of introducing the biomass into the combustion chamber and the temperature inside it (which also relates to its thermal isolation capacity) [24]. When the option is to perform $CO_2$ enrichment inside the greenhouses, the vegetal waste would require pretreatment processes to filter out these compounds [25]. This means that it would be important to optimize any prior combustion [25,26].

Previous works have studied CO, $NO_X$, and $SO_2$ emissions, combustion efficiency, and alternatives for increasing the performance of various types of biomass [27–30]. Other works have considered tomato-crop biomass for combustion [15,31–35]; the former also studied this application and implemented several alternatives for increasing its quality as a solid fuel [15]. One previous work studied the biomass obtained from tomato crops, but only that portion comprising the discarded fruit remains after juicing (tomato pomace); moreover, in this case, the combustion performance tests were for a boiler with a lower heating power output (12 kW) [36], which is not enough for use in greenhouses. There are no reports based on the combustion efficiency or toxic emissions for this biomass type (applying the pretreatments proposed in Reference [15]). Furthermore, given the importance of factors, such as the device's combustion chamber geometry, the air/fuel supply combination, and the turbulence, even if studies did exist, it would be advisable to optimize each device type for each biomass used.

Bearing in mind the above considerations, the next step is to test the combustion performance of greenhouse tomato-crop biomass in combustion devices that have a closer thermal output to those used in commercial greenhouses. This work hypothesized that combustion efficiency could be optimized, and noxious gasses minimized, by applying an appropriate combination of the previously mentioned parameters (fuel, and primary and secondary air inlets). Different combinations of these parameters were tested, and the combustion efficiency and emissions were measured to test this hypothesis. At the same

time, other more conventional types of biomass (pine pellets and olive pits) were studied to see how the influence of these parameters changes from one biomass to another.

## 2. Materials and Methods

### 2.1. Biomass

Four different biomass types were studied in this work: Olive pits, pine pellets, tomato-crop biomass, and a blend of the latter with almond prunings (75–25%). In a previous work, several characteristics were determined to study their appropriateness for use in direct combustion applications [15]. These characteristics were the water and ash content, and the calorific value. The tomato-crop biomass comprised the aerial parts of the plants, which were collected in such a way as to avoid contact with the greenhouse soil; then, they were dried, and any raffia was removed. The entire aerial portion was used (including the stems and leaves). The only part discarded was the roots. Discarding the roots significantly decreases the ash content (as demonstrated in previous research) [15]. The plants were laid out to dry for about two weeks in the same greenhouse in which they were grown. The equilibrium moisture content ranged from 10.2–15.2 (for 100% tomato and for the blend, respectively). After being dried, the biomass was chopped and pelletized. The pellet dimensions were 2.00 cm long and 0.25 cm in diameter. Additional data from the various biomass types studied are given from Figures 1–3.

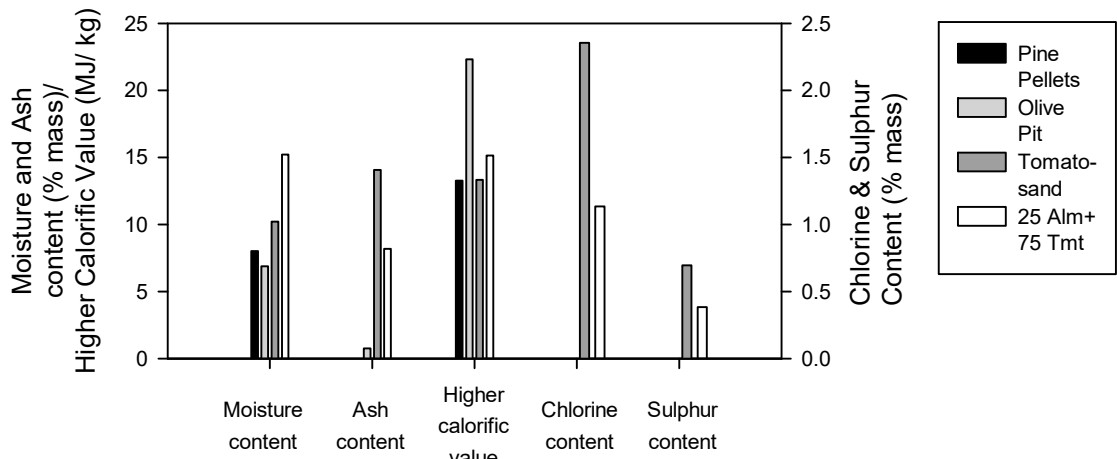

**Figure 1.** Properties of the pine pellets, olive pit, tomato biomass, and the blend of this with almond prunings.

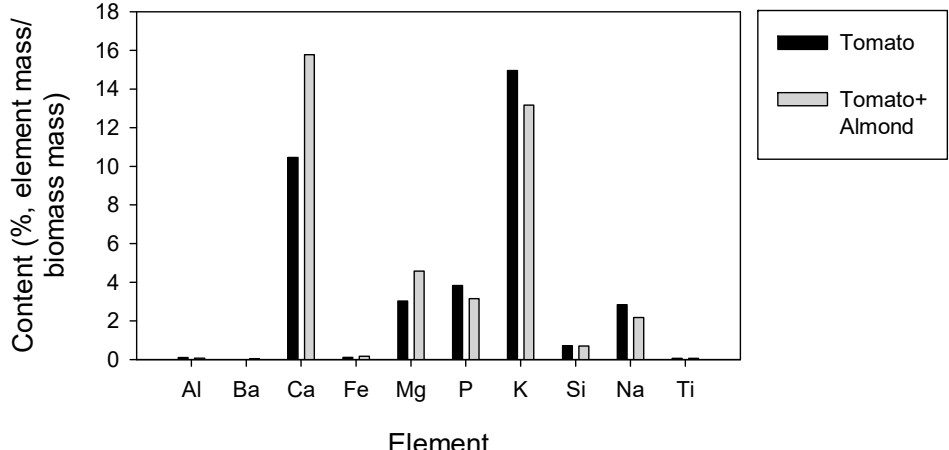

**Figure 2.** Elemental composition of the tomato-crop biomass, and the blend of this with almond prunings.

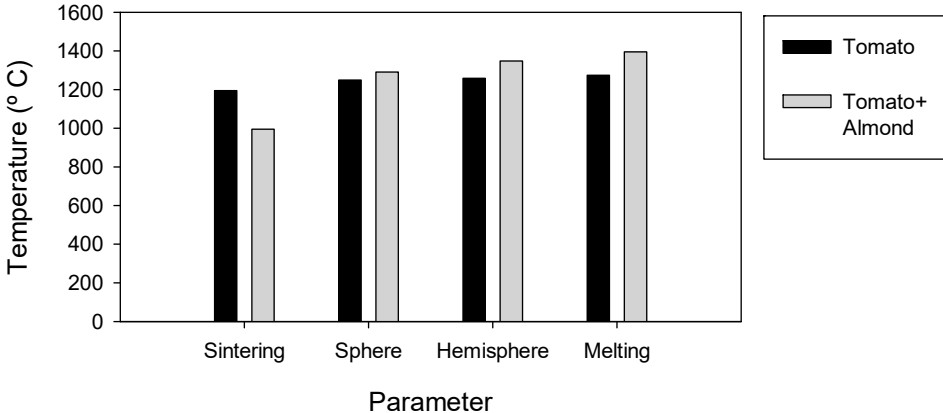

**Figure 3.** Sintering, sphere, hemisphere, and melting temperatures of tomato-crop biomass, and the blend of this biomass with almond prunings.

### 2.2. Boiler

The boiler employed for these tests was a Missouri 150000. Its nominal calorific power was 160.46 kW (Figure 4). It has a grill at its base shaped like a well. The biomass is channeled with a headless screw to the bottom of this grill and emerges from there. The flue gasses generate flow from the biomass pile through the heat exchanger. This material is placed on top of the fireplace. There are two air inlets in the combustion chamber (primary and secondary) to supply oxygen for combustion. Both air supplies are propelled with blowers. Additional details concerning boiler internal design are given in Appendix A (Appendix A.1). The primary air supply is regulated with a frequency regulator, while the secondary air supply is regulated with a flow gate. At the same time, the fuel rate can also be regulated, since it is possible to adjust this screw's rotational velocity. The $O_2$:fuel ratio can be regulated by the combined adjustment of these three parameters.

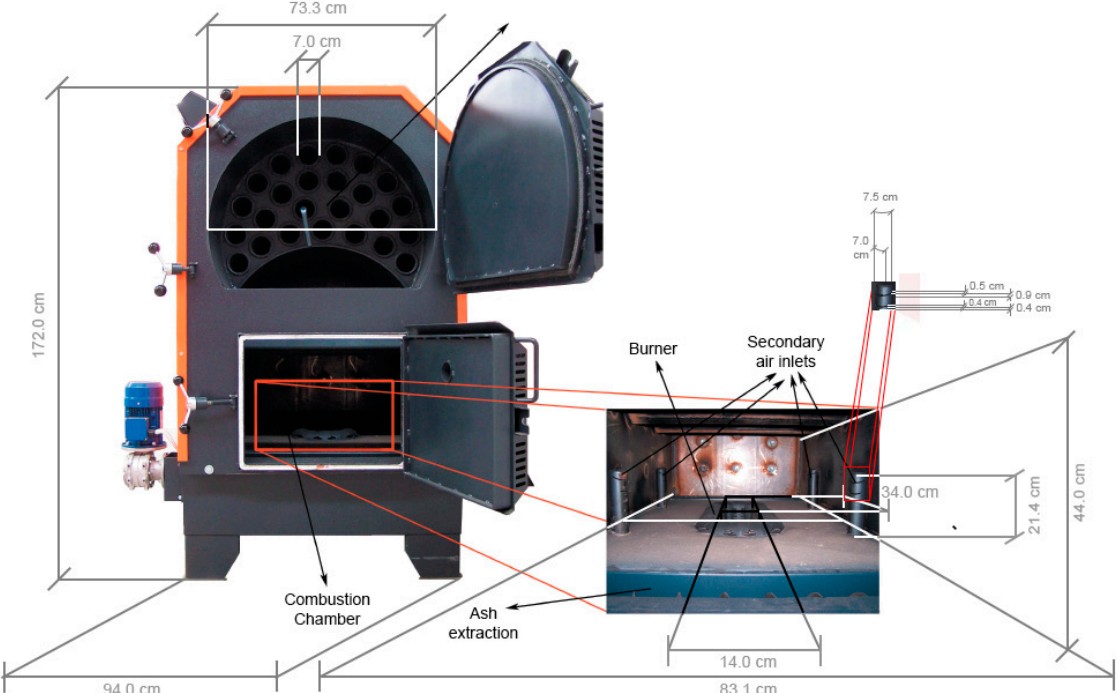

**Figure 4.** Biomass boiler, combustion chamber, and dimensions.

### 2.3. Combustion Optimization Assays

Several combinations were tested. These consisted of varying the biomass feed, and the primary and secondary air supplies. The aim was to identify the combination with the highest combustion efficiency and flue-gas $CO_2$ concentration (minimizing the excess air). The configurations tested are shown on the X-axes from Figures 5–9. These have been denoted as P#- S# (the letters "P" and "S" standing for the Primary and Secondary air supplies while the number corresponds to the flow rate assayed in $m^3 \cdot s^{-1}$). Regarding the fuel feed rate, the range selected was a bit narrower (or conservative) than the one which could actually be selected. The fuel feed rate was regulated via the rotational velocity of the headless screw adjustment (as discussed in Section 2.2.). The rate also varies from one biomass type to another, since the density of each may be slightly different. The primary and secondary air supply flow rates were estimated from experimental linear velocity measurements performed with a thermal anemometer.

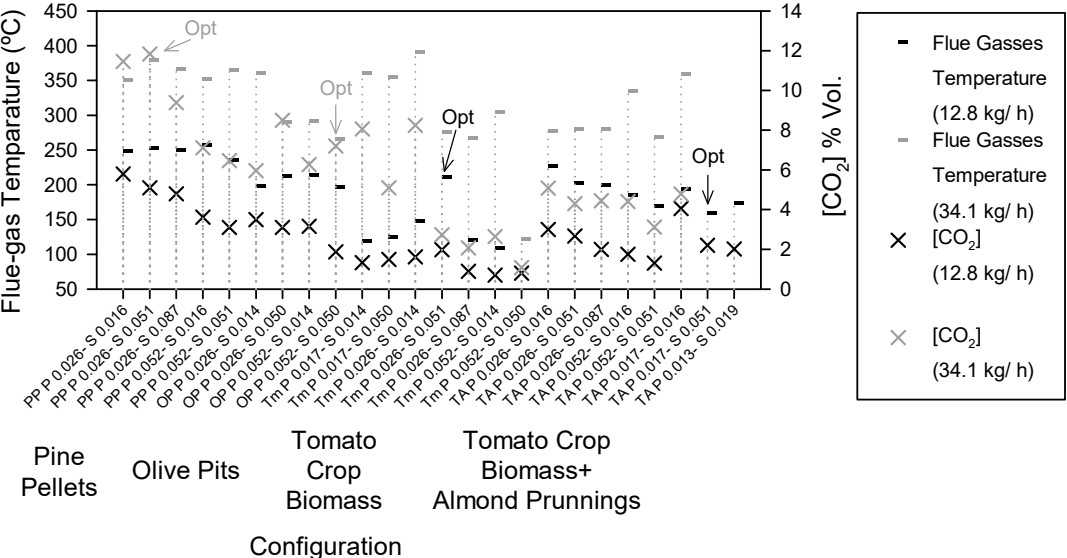

**Figure 5.** Flue-gas temperature and [$CO_2$] were recorded for each biomass type, their corresponding fuel inputs, and the primary and secondary air rates. (Opt: The configuration identified as optimum).

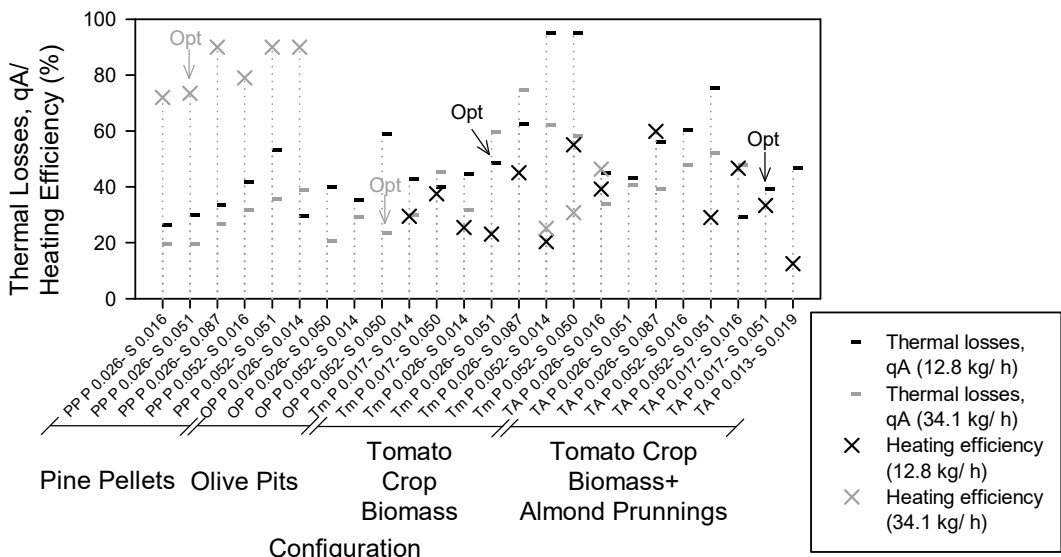

**Figure 6.** The thermal losses and heating efficiency recorded for each biomass type, their corresponding fuel inputs, and the primary and secondary air rates, together with the corresponding heating efficiency (Opt: Configuration identified as optimum; * Value surpassing 90%, theoretically).

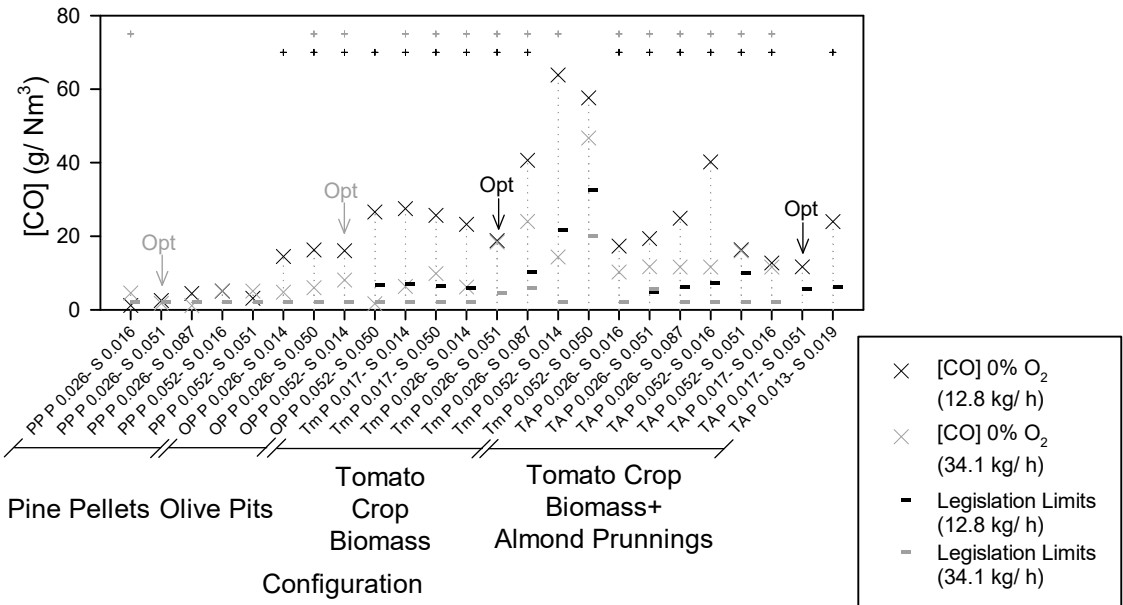

**Figure 7.** [CO] 0% $O_2$ observed for each biomass type, their corresponding fuel inputs, and the primary and secondary air supply rates (* These values surpassed the maximum that the analyzer was able to record. Opt: Configuration identified as optimum).

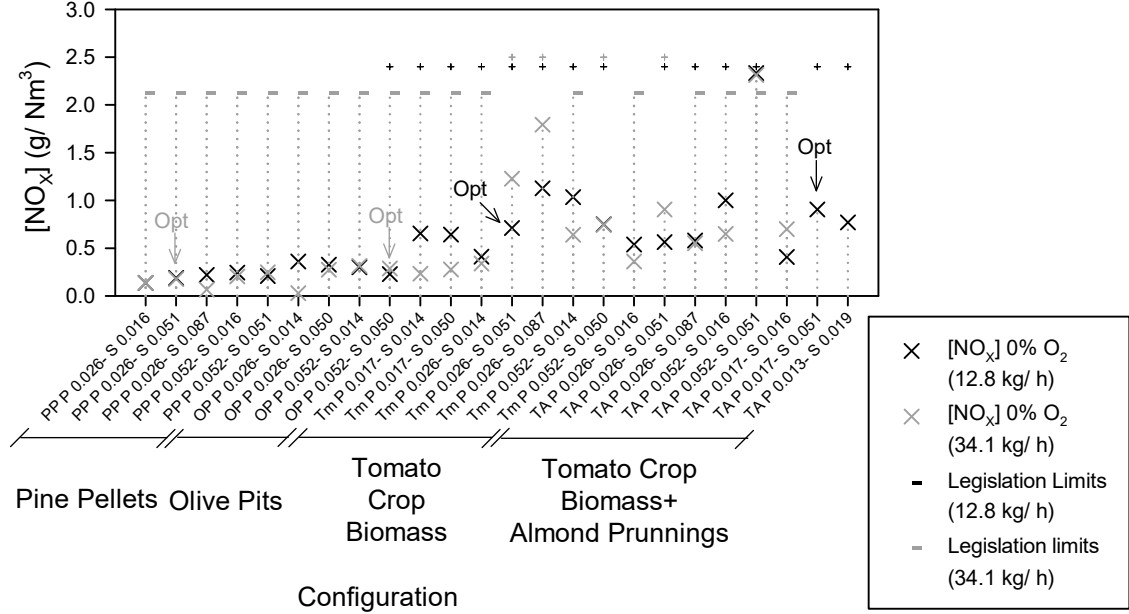

**Figure 8.** [NO$_X$] 0% $O_2$ observed for each biomass type, their corresponding fuel rates, and the primary and secondary air supply rates (*Legislation limit outside the graph scale. Opt: Configuration identified as optimum).

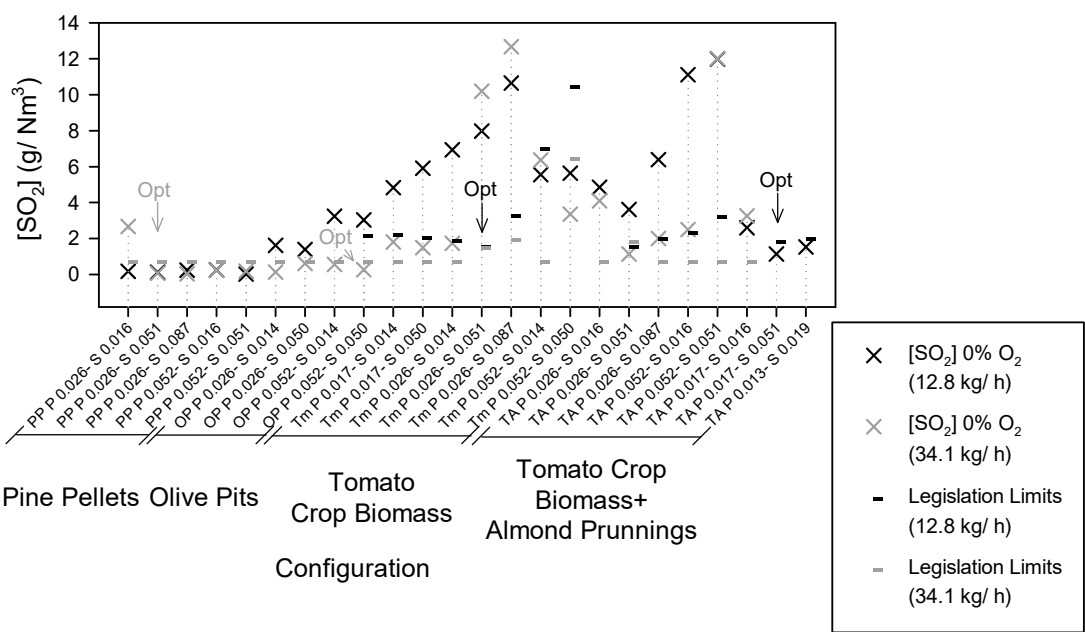

**Figure 9.** [SO$_2$] 0% O$_2$ observed for each biomass type, their corresponding fuel rates, and the primary and secondary air supply rates (*Legislation limit outside the graph scale. Opt: Configuration identified as optimum).

### 2.4. Flue-Gas Temperature and CO$_2$

These variables were measured with a gas analyzer (IM 1400 combustion analyzer, IM Environmental Equipment, Heilbronn, Germany). The analyzer's probe was placed in the boiler's flue-gas outlet pipe.

### 2.5. Thermal and Heating Efficiency Measurements

The thermal efficiency was quantified with the same analyzer mentioned in the previous section. This measurement was performed using an estimation involving the CO$_2$, O$_2$, atmospheric temperature, and flue-gas temperature. Additionally, energy and mass balances were measured to estimate heating efficiency. These were estimated for periods in which heating was taking place in the boiler, but with no water circulation. This operation makes it simpler to determine the energy transferred as one only needs to measure the water-tank temperature increment. The energy transferred can be quantified by this measurement. This estimation is further explained in Appendix B. Similarly, this estimation was performed for the pine pellets, tomato-crop residues, and the blend of tomato-crop residues + almond prunings. These assays were also performed on several set-ups out of those tested when determining the other variables under study.

### 2.6. CO, NO$_X$, and SO$_2$ Flue-Gas Measurements

These concentrations were measured with the same analyzer mentioned in Section 2.4. The CO, NO$_X$, and SO$_2$ levels observed have been recalculated hypothetically, assuming 0% O$_2$ (mol./mol.) in the flue gas. Each measurement was taken after 15 min of stable performance. The parameters analyzed were recorded for 10 s after this stabilization. An example of this procedure is given in Appendix A (Appendix A.2).

### 2.7. Posterior Reduction in CO, NO$_X$, and SO$_2$ Emissions during CO$_2$ Capture

A certain proportion of the flue gases generated is taken up for CO$_2$ capture. A tank filled with active carbon (GMI P 4 S, CPL Activated Carbons Iberia -CPL GalaQuim-, Madrid, Spain) was used for the purpose. This material has a higher adsorption capacity for CO$_2$ than for toxic gasses, increasing with pressure and decreasing with temperature. The working operational conditions were set to $2 \times 10^5$ Pa and a maximum temperature

of 40 °C. The system used is also described in References [15,33,35]. The levels present in the gas stream coming from the tank were measured on their way to the $CO_2$ enrichment process in the greenhouse (once the capture process was completed). The compounds analyzed were CO, $NO_X$, and $SO_2$. An environmental gas analyzer was used for these measurements (MultiRAE Lite; Rae Systems Spain, S. L., El Prat de Llobregat, Spain).

*2.8. Statistical Analysis.*

The influence of the considered factors (biomass type, fuel rate, and primary and secondary air supplies) has been analytically estimated. This analysis consisted of a multifactor ANOVA table. The dependent variables were those studied in this work (flue-gas temperature, $[CO_2]$, thermal efficiency, $[CO]$, $[NO_X]$, and $[SO_2]$). A maximum P-value of 0.05 was taken as a reference for a 95% confidence level. The 2nd-order iteration was considered; this gives some idea of the possible cross-over influences between these parameters. Those considered as plausible were: (1) Biomass type vs. primary air supply; (2) primary vs. secondary air supply; and (3) primary air supply vs. fuel rate.

*2.9. Particle Emissions*

Certain qualitative measurements were performed employing the Bacharach scale with the combustion analyzer mentioned previously in Section 2.4. (IM 1400 combustion analyzer; IM Environmental Equipment, Heilbronn, Germany). These measurements were performed only for the most significant configurations tested.

*2.10. Heating and Enrichment Experiments over Long Periods*

Heating and enrichment experiments were carried out inside the greenhouse where the *Solanum Lycopersicum* plants (tomato) were grown. These experiments were performed over four months, during the coldest season (from November to March), the time when the crops mentioned are usually grown in this zone, since it is possible to achieve greater productivity during this season compared to other locations. Heating was used in the greenhouse to maintain a temperature of around 10 °C. The set point was increased to 12 °C for two hours (6–8 a. m) to accelerate the plants' metabolism in the initial hours of the day. $CO_2$ enrichment was performed to maintain a $CO_2$ concentration above 1375 mg/Nm³; this only happened during periods when the greenhouse ventilation windows could be kept closed. The $CO_2$ enrichment procedure has been recommended in previous works [21,37]. These experiments were performed in two multi-span "Parral-type" greenhouses. Both have the same surface area (877 m²) and are located next to each other on the same site (Las Palmerillas Experimental Station). The cover material is polyethylene. Pine pellets were used for these experiments.

**3. Results**

The combinations are discussed in Section 2.3. were denoted as follows: P#- S#; the letters "P" and "S" stand for Primary and Secondary air supplies while the number corresponds to the flow rate assayed (in $m^3 \cdot s^{-1}$). The same configuration was tested for two different inlet fuel-rate values. The CO, $NO_X$, and $SO_2$ levels were compared with those established as the limits in the relevant legislation [38]. They were also recalculated assuming 0% $O_2$; consequently, the limits found in these figures vary compared to the configuration considered. The excess air levels employed for each configuration are related to this consideration. Additional considerations are analyzed in Appendix C.

*3.1. Flue-Gas Temperature and [CO₂]*

The data corresponding to the flue-gas temperature and $CO_2$ concentration are plotted in Figure 5.

### 3.2. Thermal and Heating Efficiency

The data corresponding to thermal and heating efficiency is plotted in Figure 6. The >95% thermal efficiency values correspond to measurements above the analyzer limit. The heating efficiency was estimated from the energy and mass balances previously described in Section 2.5. Configurations over 90% have been highlighted.

Compared with previously reported data on heating efficiency, the values observed in our experiments for the most suitable configurations using pine pellets were in a similar range to the most favorable ones reported in the literature [30]. On the other hand, the value observed for tomato pomace (fruit waste after juicing) was 91.5% (the bibliographic reference previously discussed in Section 1) [36]—higher than that observed in the present work; although that boiler had a higher thermal output.

### 3.3. [CO], $NO_X$, and $SO_2$ in the Flue Gasses

The data corresponding to the CO concentration levels is plotted in Figure 7. For some configurations, the level surpassed the maximum that the analyzer can measure (2.5 $g/Nm^3$); these values are also highlighted.

A lack of $O_2$ supply in the combustion chamber results in higher CO levels. This fact is observed in the less appropriate configurations. It has also been discussed in previous works, and is indicative of an inappropriate set-up being used [30]. On reviewing previous research, the values reported ranged between 0.6–0.8, 36.8, and 1.1 $g/Nm^3$ for pine pellets [21,36,39], olive pits [27], and tomato pomace (fruit waste after juicing) [36], respectively. Another work reported higher levels for pine pellets, although there was one extreme discrepancy (up to 923 g/Nm) from the range commented upon [40]. The range was similar, but slightly lower than the one observed in this work, apart from the discrepancy. Concerning olive pits, the minimum values observed in this work were lower than those reported in others. For tomato pomace, the values observed in the present work are considerably higher. Nevertheless, it should be noted that this study has been performed using a boiler with a higher thermal output and tomato-crop biomass from different parts of the plant.

The data for $NO_X$ concentration levels are presented in Figure 8. The analyzer maximum was surpassed for various configurations, including for this variable. The corresponding values are also highlighted in Figure 9. Previous research reported values ranging between 27–989, 411–1792, 2333, and 1662–92,206 $mg/Nm^3$, respectively, for pine pellets [21,36,39,40], tomato pomace [36], and other biomass types [28,30]. The values observed in this work were lower when pine pellets were used. Nonetheless, higher values were obtained according to the set-up. In addition, the previous work that studied tomato pomace combustion reported far higher values. For this, one should bear in mind the consideration discussed in Section 3.3. In the previous work that studied olive pits, no $NO_X$ levels were reported. In these works, the levels observed were higher than those reported for other biomass types.

The data for $SO_2$ emissions observed as flue gasses are plotted in Figure 9. On reviewing other works in the literature, $SO_2$ levels were found to range from 14–75 [27,29–31], 908 [32], and 0–3504 $mg/Nm^3$ [21–23], respectively, for pine pellets, tomato-crop biomass, and a variety of other biomass types. The lowest values compared to those reported in other works were for pine pellets. The value reported for tomato pomace was also lower than those in the present article. Nevertheless, the lowest levels observed for the blend were similar in magnitude to those in the present article. Furthermore, the considerations discussed in Section 3.3. should be noted. The work that reported CO emissions for olive pits did not report on the $SO_2$ levels. Compared to other biomass types, the values observed in the present work were within the range reported, except for the less appropriate configurations.

The results from the various works mentioned in the bibliography have been compared in Appendix D.

### 3.4. Posterior Reductions in CO, NO$_X$, and SO$_2$ Emissions during CO$_2$ Capture

Reductions of 72.09, 99.99, and 99.99% were recorded for CO, NO$_X$, and SO$_2$, respectively. This decrease was managed thanks to the filled tank containing an active carbon bed (as discussed in Section 2.7). These values relate to the relationship between the gas stream injected into the greenhouse during enrichment and the flue gases emitted from the boiler at the optimal set-ups. Concerning the 99.99% values, the actual reduction measured was 100.00%. Since this value is theoretically quite difficult to achieve, they have been corrected to 99.99% (this is a separation process in which a 100% separation yield is almost impossible).

### 3.5. Particle Emissions

The measurements taken for each biomass type ranged between 3–5, 3–6, 4–7, and 4–8 on the Bacharach scale for pine pellets, olive pits, tomato-crop residues, and the blend of the latter with almond prunings, respectively.

### 3.6. Statistical Analysis

The various *p*-values estimated for the interactions between the studied factors are plotted in Figure 10. The influence of these factors and interactions having a *p*-value lower than 0.05 could be considered statically significant at a 95.0% confidence level. This 0.05 limit value has also been plotted in Figure 10. Concerning the 2nd-level interactions, the potential interactions considered were those between the primary air supply and the fuel type. This is because the primary air stream faces towards the grate. Hence, some influence might exist. The density of the fuel pile emerging from the pit is probably different for each biomass type. This might be more important when fuel accumulates in the pit because the fuel input rate could be higher than the combustion propagation. Moreover, these two supplies are mixed inside the combustion chamber, so there is an additional interaction.

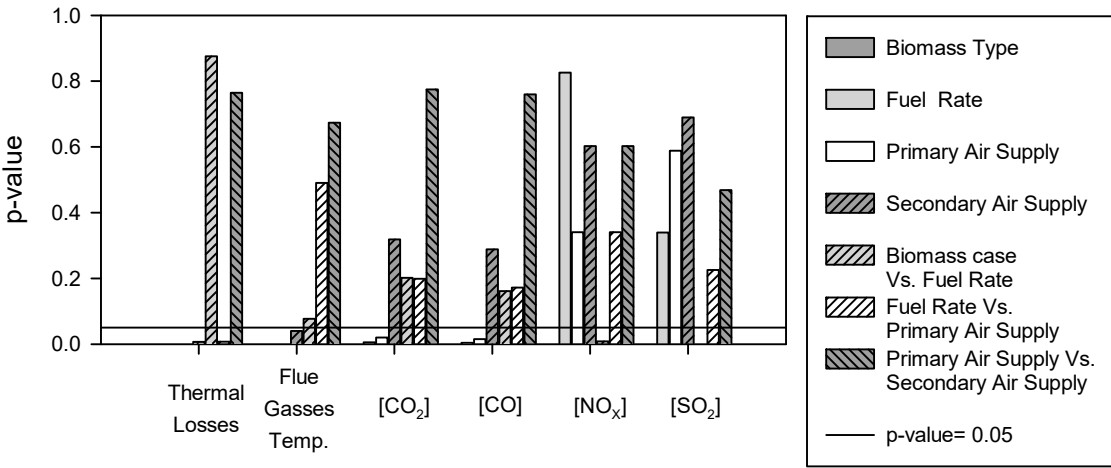

**Figure 10.** *p*-value estimated from the analysis of variance performed to study the influence of the four factors (and their combined interactions) on various variables; confidence level—95%.

### 3.7. Heating and Enrichment Experiments over the Long Term

The productivity of these crops increased by 16% (mass/mass) by the end of the growing season when following the methodology explained in Section 2.10. This increase is a result of the controlled conditions imposed on the crops and the accumulated productivity. The weekly productivity also increased by more than 10% (mass/mass over a week). This was observed three months after implementing the temperature and [CO$_2$] control regime.

## 4. Discussion

From the data obtained in the present work, one can state that the variables and parameters studied have interwoven relationships. This observation is reinforced by the data obtained from the statistical analysis (Section 3.6). This analysis showed how several interactions between two factors had a statistical significance.

Previous studies have concluded that the main contribution to the $NO_X$ and $SO_2$ levels is the nitrogen and sulfur content (as discussed in Section 1). Therefore, one would expect the emission levels to vary in line with the fuel supply rates. However, the experimental data obtained show that this relationship is not proportional. Some of these toxic gasses could remain fixed in solid compounds (for example, in ashes) [41–43]. At the same time, the other two parameters studied (the primary and secondary air rates) can influence the equilibrium of these reactions; and their combination along with the fuel rate also has some impact. Certain elements were present at higher levels in the almond-pruning biomass, namely, calcium and magnesium. It has been proved that these remain with nitrogen and sulfur. Consequently, ash minerals are generated containing these elements, resulting in lower gas-emission levels [30,44]. Additives containing these elements have been successfully assayed in another work, which showed a decrease in noxious gas emissions [45].

There are other possibilities to this mixing alternative that might provide further improvements, for example, optimizing the gas-flow patterns inside the combustion chamber [46]. On the other hand, there are parameters, such as the watering rate and the salinity content of the soil in which the plants grow, which influence the structural composition of the biomass. This influence might be more significant in the final days leading up to plant removal. Likewise, controlling the growing parameters during the final days or weeks before cutting the plants down can influence the structural and elemental composition of the biomass. In this regard, some research has reported that the elemental and structural composition of energy crops fluctuates depending on the season and location [47,48]. Additionally, other research has shown that the soil salinity present where the crops are grown can even influence nitrate and phosphate transport (and thus, the biomass' chemical composition as well) [49]. Pretreatment measures, such as washing, have also been introduced; these were able to decrease the ash-forming elements in other types of biomass. Researchers have successfully tried this technique with crop residues from pepper plants [50]. That work also reported a decrease in the chlorine (Cl) content. The same technique was tried on tomato-crop biomass, leading to a 12% reduction in the ash content [32]. Complementary to these alternatives, there is a report of micro-fungi being used to treat other types of biomass (sugar cane bagasse and rice husks), which fixes a part of these elements and results in some ash-content reduction, particularly the Cl content, which is quite interesting in terms of solid fuels [51].

There were two possible criteria for this optimization (1) $CO_2$ concentration or (2) thermal combustion efficiency. The second was considered the most appropriate for the present work. Additionally, it was important that the CO, $NO_X$, and $SO_2$ emission levels did not exceed regional legislation limits [38]. Regarding the tomato-crop biomass (100%) and the blended biomass, unacceptable combustion performance was observed for some of the experimental configurations. This was reinforced by the corresponding flue-gas temperature and $CO_2$ levels recorded, variables that were also considered for these biomasses.

The optimal fuel-rate configurations were 34.1, 34.1, 12.8, and 34.1 kg/h for the pine pellets, olive pits, tomato-crop (100%), and blended biomass, respectively. The optimal primary air supply was 0.026, 0.052, 0.026, and 0.026 $m^3$/s for the same respective biomass order as previously stated. The optimal secondary air supply was 0.051, 0.051, 0.016, and 0.087 $m^3$/s. These optimal set-ups are also highlighted in Figures 5–9. It was observed that a slightly lower efficiency with lower fuel rate input uses the same configuration for the primary and secondary air supply. However, these configurations would be more convenient as they would prevent the boiler from being activated too frequently during

periods when the thermal requirements are lower than 100%. Furthermore, overly-frequent ignitions/stops increase fuel consumption and noxious gas emissions [20]. For some biomasses, the optimal set-up corresponded to the maximum fuel input, a point worth considering.

## 5. Conclusions

From the experimental data, it can be stated that combining the three factors under study (primary and secondary air, and fuel supply) significantly impacts combustion performance. In several cases, the tendency of one of these factors varied according to the combination of the other two. The role of $O_2$ on C can be highlighted considering the interaction between these factors. This has an impact both on combustion performance and on toxic-gas emissions, as supported by the *p*-values estimated from the statistical analysis we performed.

In addition, the boiler used in this work is designed mainly for commercial biomass combustion (pine pellets or olive pits being the most widely used). Consequently, variations in the combustion chamber design could be tested, which might increase the combustion performance of the tomato-crop pellets (or the blend with almond prunings). These variations could focus on the flow patterns of the flue gasses and/or the thermal isolation capacity of the combustion chamber.

Selecting the proper boiler set-up is essential for optimizing the combustion performance of each biomass type. For example, a comparison between the least and most favorable set-ups found increases of 63.3, 65.2, 58.2, and 55.2% in thermal losses; 87.3, 87.5, 77.5, and 74.5% in CO emissions; 88.0, 89.0, and 77.1% in $NO_X$ emissions; and 99.3, 95.5, and 62.4% in $SO_2$ emissions for pine pellets, olive pits, tomato-crop biomass, and the latter blended with almond prunings, respectively. These data refer to values re-estimated for 15% $O_2$. Moreover, these were not the only increments observed—various configurations had emission levels higher than those permitted by the relevant regional legislation [38].

Optimizing combustion performance is important for reducing fuel consumption, which is relevant when it comes to lowering the cost of fuel for heating. At the same time, the reduction in toxic-gas emissions is important for subsequent $CO_2$ capture. This could be captured from the $CO_2$ generated from biomass combustion. Filtering processes would be necessary to lower these toxic-gas emissions. The captured $CO_2$ would then be used for enrichment. Combining heating and $CO_2$ enrichment has productivity benefits for the crops grown in the greenhouse.

It should be noted that the optimal set-up for each biomass was slightly different. The combustion efficiency of the tomato-crop/almond pruning biomass blend was slightly better than the others tested, but the toxic-gas emissions were higher. The optimal set-up for both these variables was comparable to those for the 100% tomato-crop biomass. This observation is similar to that for the combustion properties (Section 2.1. and in previous bibliographical work [15]). The elemental composition capable of fixing $NO_X$ and $SO_2$ has an influence on the corresponding emissions, as explained in Sections 1 and 3.3. By mixing in almond prunings, it is possible to increase the proportion of these elements. With this consideration in mind, two biomasses with different combustibility qualities were mixed. As a result of this mixing, it was possible to increase the combustion performance of the less suitable source. Moreover, the main combustion properties of this biomass could be improved with additional pretreatments, such as washing with water.

The development of this alternative could provide an attractive option for increasing the profitability and reducing the environmental impact of the intensive horticultural activity carried out in the area that the study focused on. Nevertheless, it could be applicable in other areas where similar agricultural practices take place. This option also offers a novel alternative for revalorizing other crop residues with similar properties in applications that require thermal energy or a $CO_2$ supply, provided they are located near the sites where the agricultural residues are generated.

## 6. Patents

The patent "Combined system of heating and carbon enrichment from biomass" (ES2514090) is related to this article.

**Author Contributions:** Conceptualization, F.G.A.F.; methodology, F.G.A.F. and J.V.R.M.; software, J.A.S.M. (software responsible for the system control); validation, F.G.A.F., M.G.P.H., J.C.L.H., and J.A.S.M.; formal analysis, all the authors; research, F.G.A.F. and J.V.R.M.; resources, F.G.A.F.; data curation, J.V.R.M.; writing—original draft preparation, J.V.R.M.; writing—review and editing, M.G.P.H. and F.G.A.F.; visualization, M.G.P.H., F.G.A.F., J.C.L.H., J.A.S.M., and M.D.F.F.; supervision, F.G.A.F., M.G.P.H. and J.C.L.H.; project administration, F.G.A.F.; funding acquisition, F.G.A.F. All authors have read and agreed to the published version of the manuscript.

**Funding:** This work is part of the "Carbon4Green" research project (UAL18-TEP-A055-B) within the FEDER Operative Program for Andalusia (2014–2020) framework of the Economy, Knowledge, and Companies Office of the Junta de Andalucía (Andalusian Regional Government).

**Institutional Review Board Statement:** Not applicable.

**Informed Consent Statement:** Not applicable.

**Data Availability Statement:** The data presented in this study are openly available in the Agronomy Journal (ISSN: 2073-4395).

**Acknowledgments:** The authors of this work are most grateful to the Chemical Engineering Department of the University of Almeria, where the research work developed for this article was conducted, and to the invaluable contribution made by the "Las Palmerillas" Research Station of the Cajamar/Caja Rural Foundation.

**Conflicts of Interest:** The authors declare no conflict of interest.

## Appendix A. Methodology, Additional Considerations

*Appendix A.1. Boiler Internal Design*

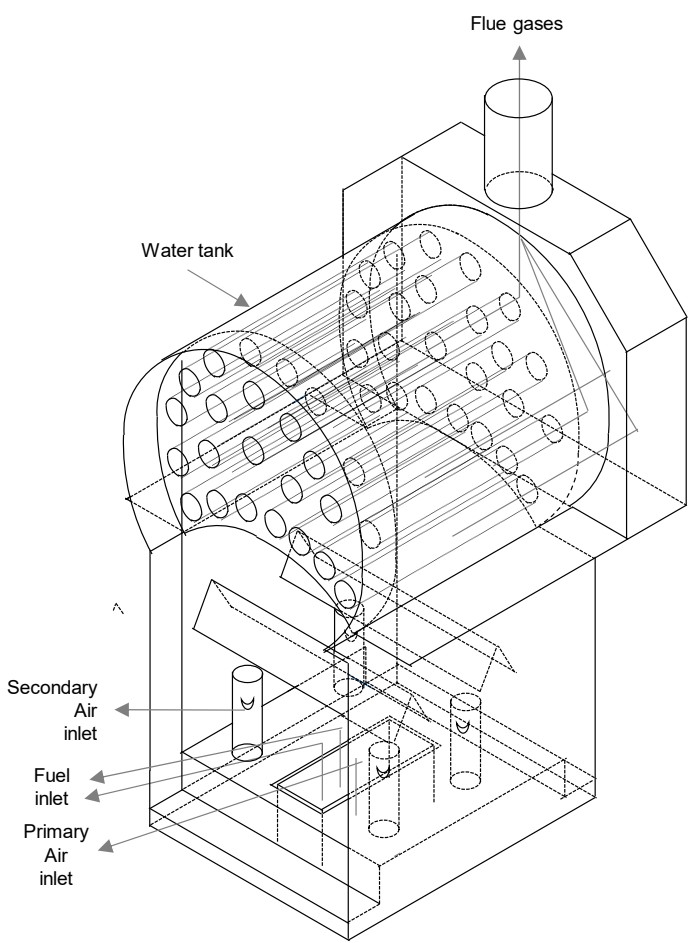

**Figure A1.** Schematic of the boiler's internal design.

*Appendix A.2. Methods Applied for the Flue-Gas Measurements (Thermal Efficiency, $CO_2$, $CO$, $NO_X$, and $SO_2$)*

Each measurement was taken after 15 min of stable working conditions. The parameters analyzed were recorded for 5 min following the stable operation being achieved. An example of a $CO_2$ measurement has been plotted in Figure A2.

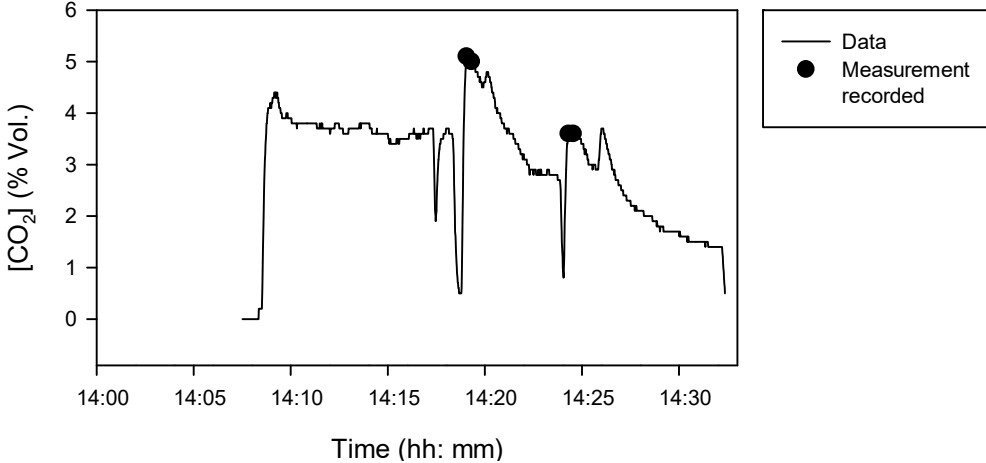

**Figure A2.** Example of the methodology followed for the measurements (example: $[CO_2]$).

## Appendix B. Heating Efficiency Estimation

*Appendix B.1. Discontinuous Experiments*

Heating is performed by means of a water circuit. The various pipes for this circuit are placed alongside the plants in the greenhouse. Figure A3 is a diagram corresponding to this circuit. Water is pumped from the boiler through the circuit. The aim is to estimate the combustion efficiency for the various configurations and biomass types. First, it is necessary to quantify the heat transferred to the water in the circuit. A simpler method would be to consider a mass/energy balance for periods when combustion takes place, and only the water volume present in the boiler's water tank is heated (with no water pumping). Hence, the generated heat will only increase the temperature of this water (apart from the thermal losses). The system efficiency can be estimated from the heat transference and the amount of fuel combusted, using Equation (A1) for this calculation; where $m_{H2O}$ is the mass of water (in the boiler's water tank); $c_{p\ H2O}$ is the water's calorific capacity; $\Delta$Temp is the temperature increment given at the beginning and end of the period considered for this balance; "$m_{fuel}$" is the mass of fuel supplied during this period; and HHV is the Higher Heating Value.

$$h = m_{H2O} \cdot c_{pH2O} \cdot \Delta Temp/(m_{fuel} \cdot HHV) \tag{A1}$$

As soon as the heating begins, so does the pumping of the water through the circuit. The water temperature evolution observed after one of these activations is plotted in Figure A4. Here, the previously explained methodology was applied before starting the water pump. One can observe several temperature fluctuations (relative maximum and minimum values) for several minutes post-activation. From this observation, one can state that the system behaves like a plug-flow reactor. The time elapsed between activation and the first maximum observed at the boiler outlet (or beginning of the circuit) is useful for estimating the flow of the recirculated stream (Equation (A2); the boiler tank volume is known). On the other hand, it is possible to observe some delay between the first temperature maximum at the beginning and end of this circuit. This time lag can be used to estimate the greenhouse's heating circuit volume with the previously estimated flow rate. The change in the flue-gas temperature during this experiment is also plotted in Figure A5. This variable is useful for checking the combustion status and stability.

$$q = V/t \tag{A2}$$

Equation (A2)—q: flow; V: boiler deposit volume; t: time elapsed until observing the relative maximum temperature.

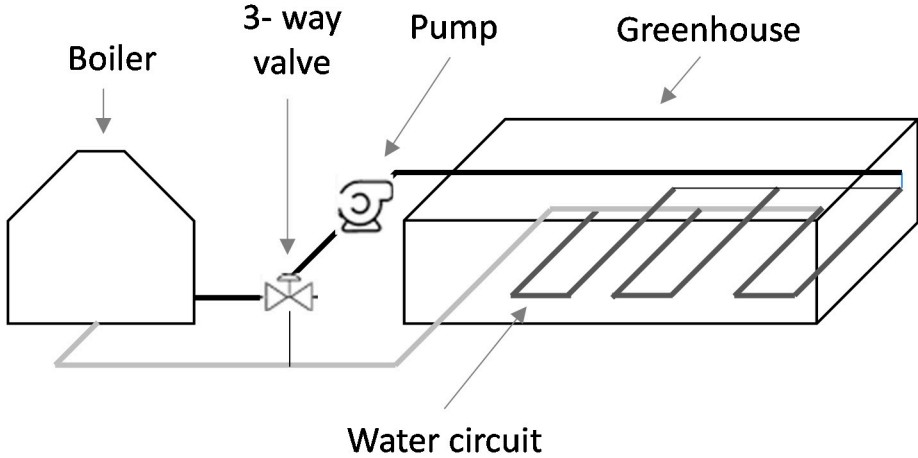

**Figure A3.** Diagram of the water circuit used for heating.

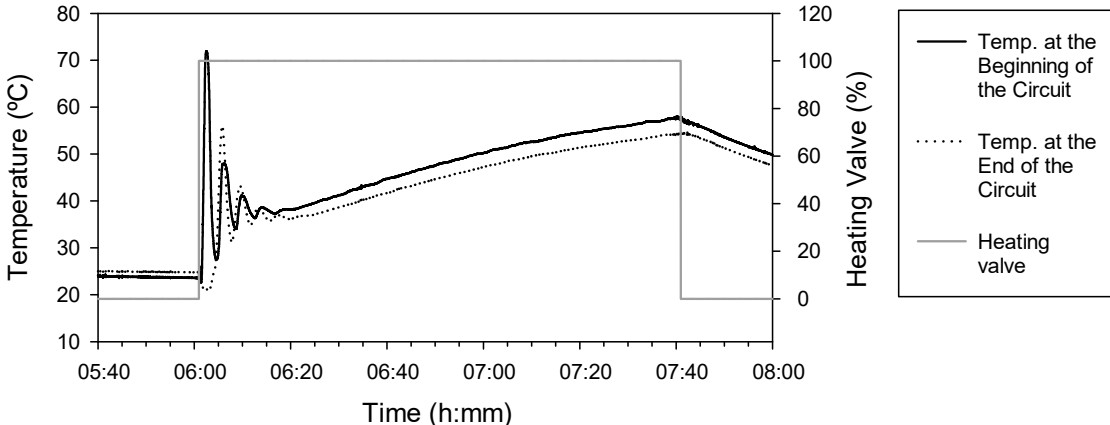

**Figure A4.** Water temperature at the beginning and end of the greenhouse circuit over a period when some biomass combustion was performed. The heating valve has also been plotted to identify the precise time when the heating was performed.

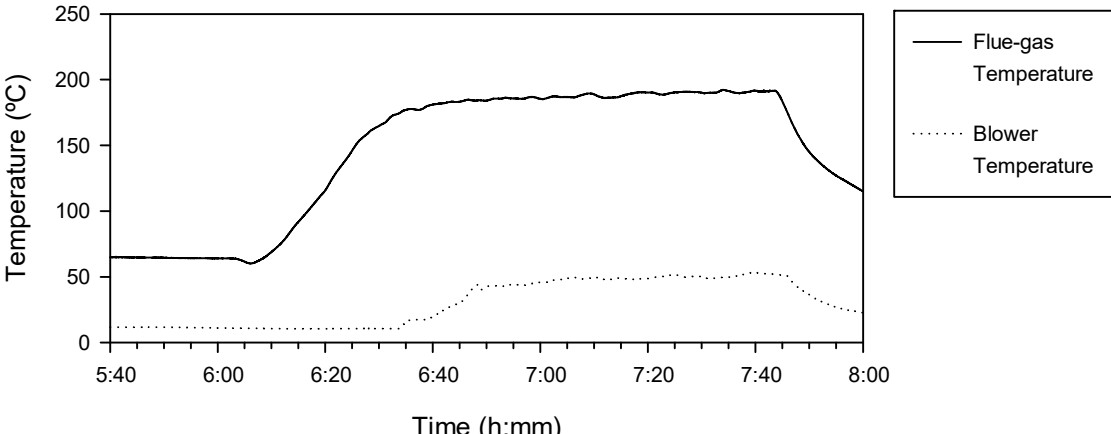

**Figure A5.** Flue-gas temperature over the period corresponding to Figure A4.

*Appendix B.2. Estimation of the Combustion Efficiency at Any Moment*

The boiler does not usually operate in the way explained in Appendix B.1. It is possible to make a similar calculation for estimating the heat transferred to the interior greenhouse environment under regular working conditions (when the water is pumped through this circuit). Another experiment was performed to do this. First, the water contained in the boiler's water tank was heated to a set-point temperature. There is a three-way valve in this circuit, so it is possible to recirculate the water without it being pumped through the boiler. The circuit can be filled with hot water. After this, the delivery of the heat supply can be avoided while allowing the hot water to recirculate through the circuit for some time. This constitutes a pulse test. The temperature change of the stream coming off the tank over time is plotted in Figure A6, following the methodology explained above. The energy transference can be calculated because the system volume is known (as explained previously in Appendix B.1.), and the temperature decrease is recorded.

Additionally, the energy transference was estimated for different water temperatures. As expected, energy transference varies with temperature because it also depends on the temperature gradient. The energy transference estimated with different water-circuit temperature values is plotted in Figure A7, where "qgiven" is the energy transference from the water circuit to the greenhouse environment and "Bg Temp" is the temperature at the beginning of the circuit. The $r^2$ was quite low. Nevertheless, some linear dependence can still be identified. The first values observed after starting the experiment were discarded

for this regression. The system behaves like a plug-flow reactor (as discussed before). Because of this, considering these values as representative of the rest of the water circuit is not recommended. The energy transference was −2.85 kW for the highest temperature observed, with the entire water-circuit volume stabilized at 25 °C. This value was the maximum possible following the methodology described above.

The normal working conditions are between 40–60 °C. These values are higher than the maximum consideration for the estimated regression (25 °C). The water temperature recorded was 56.42 °C, i.e., with on-demand heating and operating under regular working conditions. The heat transference at this temperature was −21.00 kW (from the water circuit to the greenhouse, estimated using this extrapolation). The data corresponding to the water-circuit temperature during the experiments for this estimation is plotted in Figure A8. Meanwhile, the thermal efficiency was 42.44% at this water temperature (the configuration considered optimal for pine pellets). From this, a relatively important extrapolation was made. The temperature value was quite different from the higher one considered for the linear regression. One could repeat this experiment in an attempt to achieve temperature values closer to the working conditions. A possible alternative would be to heat the entire circuit volume with hot water (both the boiler tank and the greenhouse circuit) then start pumping the water only through the greenhouse circuit (avoiding the water pumping through the boiler tank). Higher water temperatures should be achieved with this alternative methodology.

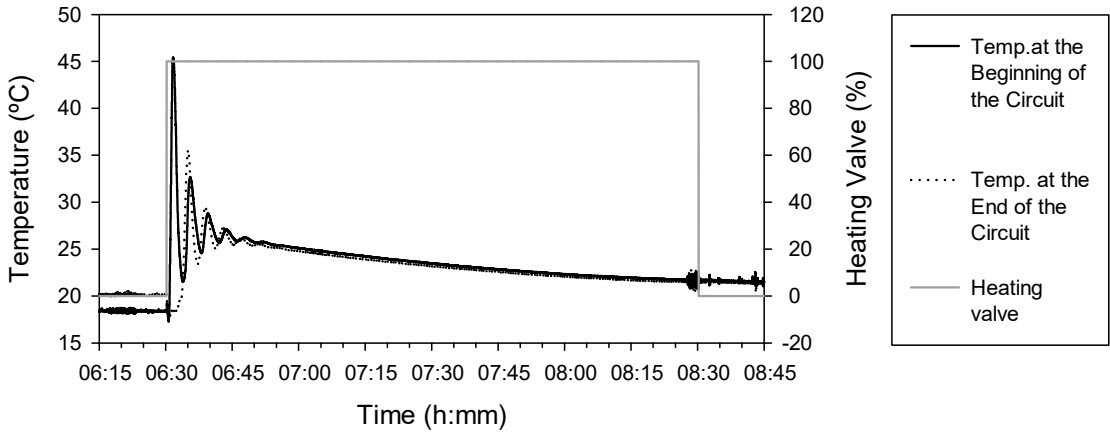

**Figure A6.** Experiment for estimating the energy transference from the water circuit to the greenhouse environment. The temperature change at the beginning and at the end of this circuit, and the heating valve status (100% implies that the heating is on-demand).

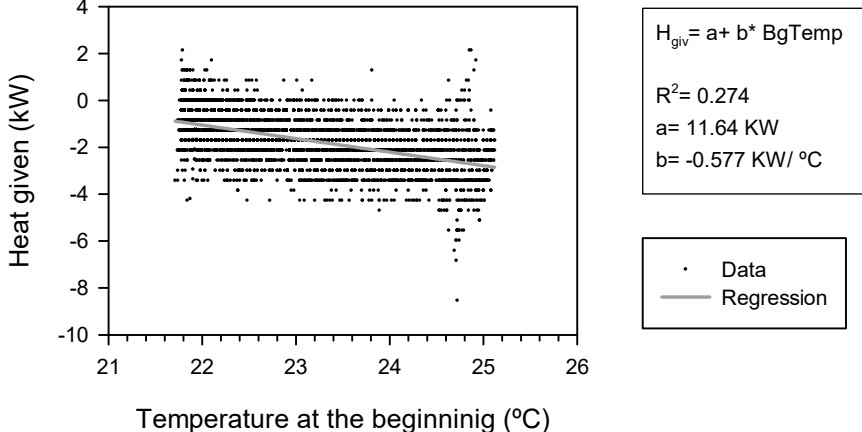

**Figure A7.** Correlation between the energy transference from the greenhouse circuit to the environment, and the temperature at the beginning of the water circuit.

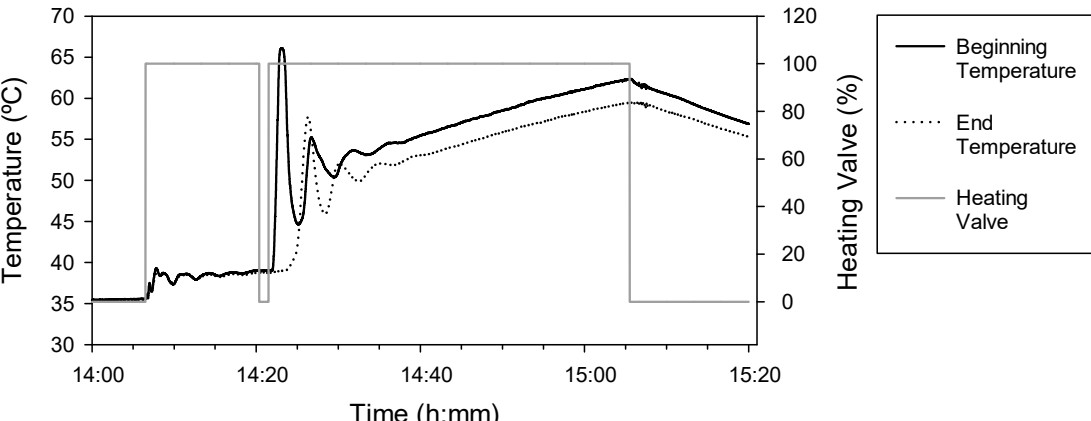

**Figure A8.** A heating experiment to estimate the combustion efficiency using continuous data (considering also the heat transference from the greenhouse circuit to its environment).

## Appendix C. Results and Discussion—Additional Considerations

### Appendix C.1. Flue-Gas Temperature and [$CO_2$]

The values observed for these parameters are presented in Figure 5. Focusing on each biomass type, the combination of air/fuel rates and turbulence had an important impact on this variable. The fuel rate is the clearest relationship observed, the higher this rate, the higher the temperature. Regarding the primary air supply, the temperature increased along with this factor. The secondary air supply was not so important; this factor's impact was less on some configurations.

The highest values observed were for the same configurations using pine pellets. There were particularly low values for some configurations, as was the case using the tomato-crop biomass, and the blend of this with almond prunings. Here, the main stages of combustion might have been dehydration and/or devolatilization (the stages occurring at lower temperatures). The secondary air supply had more impact on the configurations for these two biomasses.

With regard to $CO_2$, the values observed for these parameters are presented in Figure 5. Here, there seems to be a relationship between the fuel supply and the $CO_2$ concentration: the higher the first is, the higher the second. This increase was more evident in the case of pine pellets and olive pits. Conversely, this relationship was less clear for the other two biomasses. Concerning the primary air supply, a relationship was only observed for pine pellets. The higher this rate, the higher the $CO_2$ concentration. Regarding the secondary air supply, this was the most important impact observed for pine pellets for every set-up tested. This tendency was only observed for the lowest primary airflow using the blend composed of tomato-crop residues and almond prunings. In this case, the $CO_2$ concentration decreased slightly along with the secondary air supply rate. The $CO_2$ levels were higher when using pine pellets for all four cases tested, except for a few set-ups tested with olive pits.

### Appendix C.2. Thermal and Heating Efficiency

The values observed for these parameters are presented in Figure 6. Concerning thermal efficiency, a certain tendency was observed for this variable to increase along with the fuel rate. The tendency was clearer in the case of pine pellets and olive pits. It was also more evident for those configurations with a lower primary air supply. Conversely, thermal losses increased with this factor when maintaining the fuel rate and secondary air supply. This trend was more marked in the case of the tomato-crop biomass and the blend of this with almond prunings. The secondary air supply had a varying influence depending on the other parameters and the biomass used. The clearest influence was that observed for pine pellets and olive pits, especially with the lower fuel feed rates. Some

influence was observed for the tomato-crop biomass and the blend. The most evident influence corresponded to the lowest air supply and fuel rate.

Regarding heating efficiency, those configurations with values higher than 90% were not considered. This is because it is not plausible to surpass this value under normal conditions. The range observed was quite broad. The values corresponding to pine pellets were the highest. The values observed for the blend were slightly higher than those for the 100% tomato-crop biomass (yet significantly lower than those corresponding to the pine pellets). Conversely, the relationship between the heat transference and the thermal gradient for each biomass type, and the configuration tested, could be interesting (the transference from the flue gas to the water in the tank).

*Appendix C.3. [CO], [NO$_X$], and [SO$_2$] in the Flue Gases*

The CO, NO$_X$, and SO$_2$ emission data is presented in Figures 7–9. Concerning CO emissions, a moderate influence was observed when using pine pellets as the fuel supply. The lowest values were obtained at the highest fuel rate. This same trend was observed for the blend, although, in this case, the difference was more significant. Likewise, for the other biomasses studied, the only configurations that did not surpass the maximum were observed at the higher fuel rate. Regarding the primary air supply, a less important influence was observed with pine pellets. Its influence was more important for the blend, although this also depended on the other parameters. The configurations for which the values did not surpass the analyzer maximum at the lowest primary air supply were for the olive-pit and tomato-crop (100%) biomasses. The secondary air supply had a varying influence depending on the other parameters. Its influence was relatively stronger in the case of the blend.

Regarding NO$_X$ emissions, the fuel supply rate had an almost negligible influence, and was only barely significant in the case of the blend, where a moderate increase was observed at the highest fuel supply. The influence of the primary air supply was also moderate, a little more important for the tomato-crop biomass (100%) and the blend. The secondary air supply did affect the NO$_X$ levels in some cases. Nonetheless, this influence varied depending on the other parameters. As in the case of CO, the legislative limit values varied depending on the configuration tested because these are estimated considering 0% O$_2$. The levels observed for pine pellets were the lowest, while the corresponding levels observed for olive pits were slightly higher. Conversely, and depending on the set-up, higher levels were obtained for the tomato-crop biomass (100%) and for the blend. Furthermore, some of the lower values observed for the blend were in a similar range to some of the highest values observed for olive pits.

Regarding SO$_2$ emissions, the fuel supply rate barely had an influence. Slightly lower values were observed at the highest fuel supply for the tomato-crop biomass and the blend. In terms of the primary and secondary air supplies, their influence depended on the other parameters. Despite this dependence being variable, the difference observed between the highest and lower levels obtained was quite important.

*Appendix C.4. Statistical Analysis*

The ANOVA tables estimated for the various parameters studied (flue-gas temperature, thermal efficiency, [CO], [NO$_X$], and [SO$_2$]) are combined in Table A1. Only those factors that have a relevant statistical influence (*p*-value< 0.05) are included in this table.

**Table A1.** Analysis of Variance for the variables studied (Flue-gas temperature, thermal efficiency, [CO], [NO$_X$], and [SO$_2$])-Type III Sums of Squares.

| Parameter | Source | Sum of Squares | Df | Mean Square | F-Ratio | P-Value |
|---|---|---|---|---|---|---|
| Flue-gas temperature | MAIN EFFECTS | | | | | |
| | A: Biomass type | 4324.19 | 3 | 1441.4 | 17.73 | 0.0000 |
| | B: Fuel rate | 796.175 | 1 | 796.175 | 9.79 | 0.0053 |
| | C: Primary air supply | 516.884 | 1 | 516.884 | 6.36 | 0.0203 |
| | RESIDUAL | 1626.22 | 20 | 81.3111 | | |
| | TOTAL (CORRECTED) | 7910.74 | 31 | | | |
| Thermal efficiency | MAIN EFFECTS | | | | | |
| | A: Biomass type | 6796.97 | 3 | 2265.66 | 41.48 | 0.0000 |
| | B: Fuel rate | 1006.88 | 1 | 1006.88 | 18.43 | 0.0004 |
| | C: Primary air supply | 2040.01 | 1 | 2040.01 | 37.35 | 0.0000 |
| | D: Secondary air supply | 500.07 | 1 | 500.07 | 9.16 | 0.0067 |
| | INTERACTIONS | | | | | |
| | BC | 487.5 | 1 | 487.5 | 8.92 | 0.0073 |
| | RESIDUAL | 1092.45 | 20 | 54.6224 | | |
| | TOTAL (CORRECTED) | 11,966.3 | 31 | | | |
| [CO$_2$] | MAIN EFFECTS | | | | | |
| | A: Biomass type | 120.535 | 3 | 40.1783 | 29.95 | 0.0000 |
| | B: Fuel rate | 71.4013 | 1 | 71.4013 | 53.22 | 0.0000 |
| | C: Primary air supply | 18.0 | 1 | 18.0 | 13.42 | 0.0015 |
| | D: Secondary air supply | 6.48 | 1 | 6.48 | 4.83 | 0.0399 |
| | RESIDUAL | 26.8325 | 20 | 1.34163 | | |
| | TOTAL (CORRECTED) | 254.82 | 31 | | | |
| [CO] | MAIN EFFECTS | | | | | |
| | A: Biomass type | 4324.19 | 3 | 1441.4 | 20.41 | 0.0000 |
| | B: Fuel rate | 796.175 | 1 | 796.175 | 11.27 | 0.0037 |
| | C: Primary air supply | 516.884 | 1 | 516.884 | 7.32 | 0.0150 |
| | RESIDUAL | 1200.87 | 17 | 70.6393 | | |
| | TOTAL (CORRECTED) | 7910.74 | 31 | | | |
| [NO$_X$] | MAIN EFFECTS | | | | | |
| | A: Biomass type | 5.05303 | 3 | 1.68434 | 12.38 | 0.0002 |
| | D: Secondary air supply | 0.618828 | 1 | 0.618828 | 4.55 | 0.0478 |
| | INTERACTIONS | | | | | |
| | AC | 2.20263 | 3 | 0.73421 | 5.40 | 0.0085 |
| | RESIDUAL | 2.31231 | 17 | 0.136018 | | |
| | TOTAL (CORRECTED) | 10.588 | 31 | | | |
| [SO$_2$] | MAIN EFFECTS | | | | | |
| | A: Biomass type | 304.908 | 3 | 101.636 | 24.59 | 0.0000 |
| | INTERACTIONS | | | | | |
| | AC | 127.58 | 3 | 42.5268 | 10.29 | 0.0003 |
| | RESIDUAL | 82.6784 | 20 | 4.13392 | | |
| | TOTAL (CORRECTED) | 529.772 | 31 | | | |

Only the values relevant to factors with statistical influence have been included (*p*-Value < 0.05). All F-ratios are based on the residual mean square error.

The optimal configurations (combinations between the fuel rate, and primary and secondary air supplies) found for each biomass type studied have been collected in Figure A9. The variables observed for these configurations are collected in Figures A10 and A11.

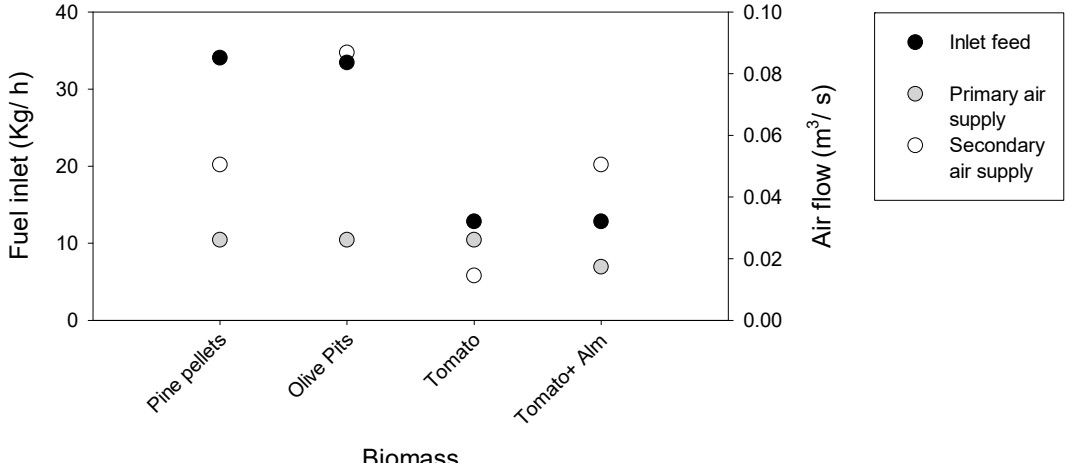

**Figure A9.** Configurations for which the best combustion performance was found for each biomass type studied.

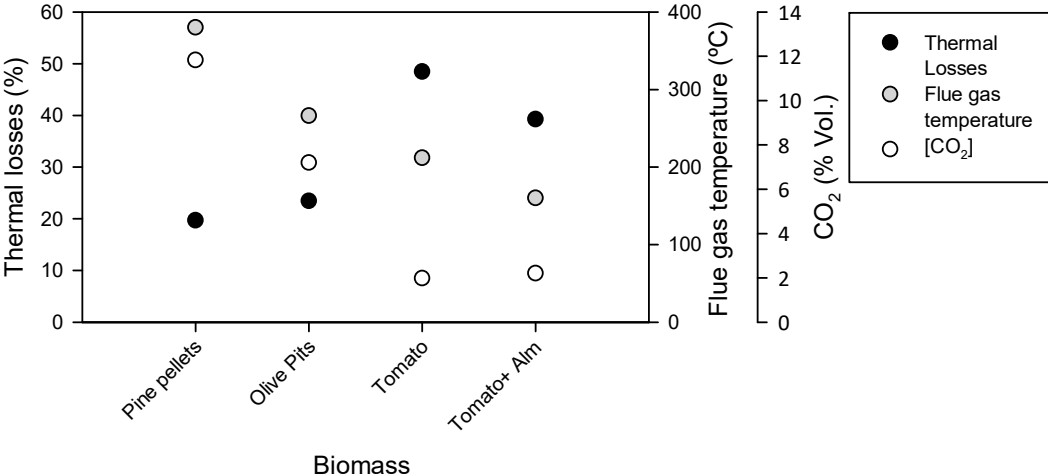

**Figure A10.** Thermal losses, flue-gas temperature, and $CO_2$ using the optimal configuration proposed.

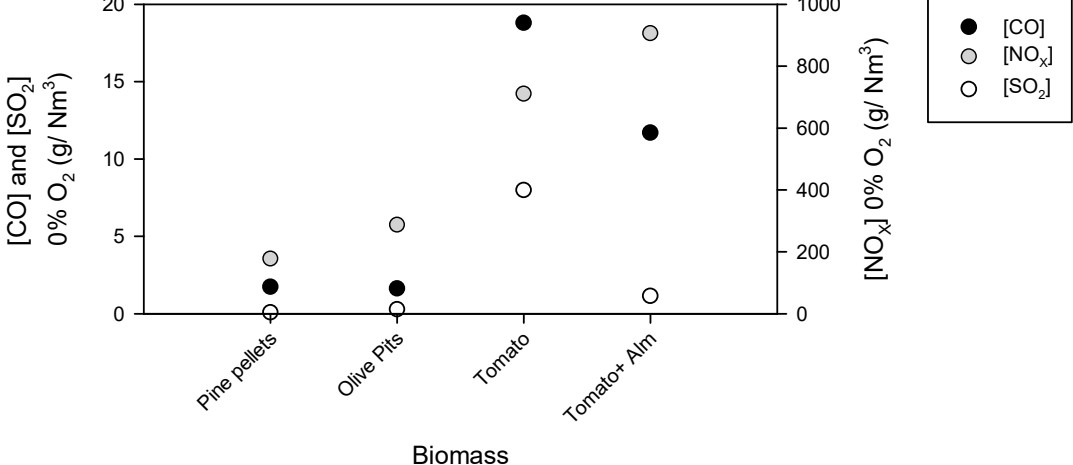

**Figure A11.** CO, $NO_X$, and $SO_2$ levels achieved when using the optimal configuration proposed for each biomass studied.

## Appendix D. Graphical Comparison with the Bibliography Results

The results from various works in the bibliography have been collected and plotted together. These values are collected in Figure A12, Tables A2 and A3.

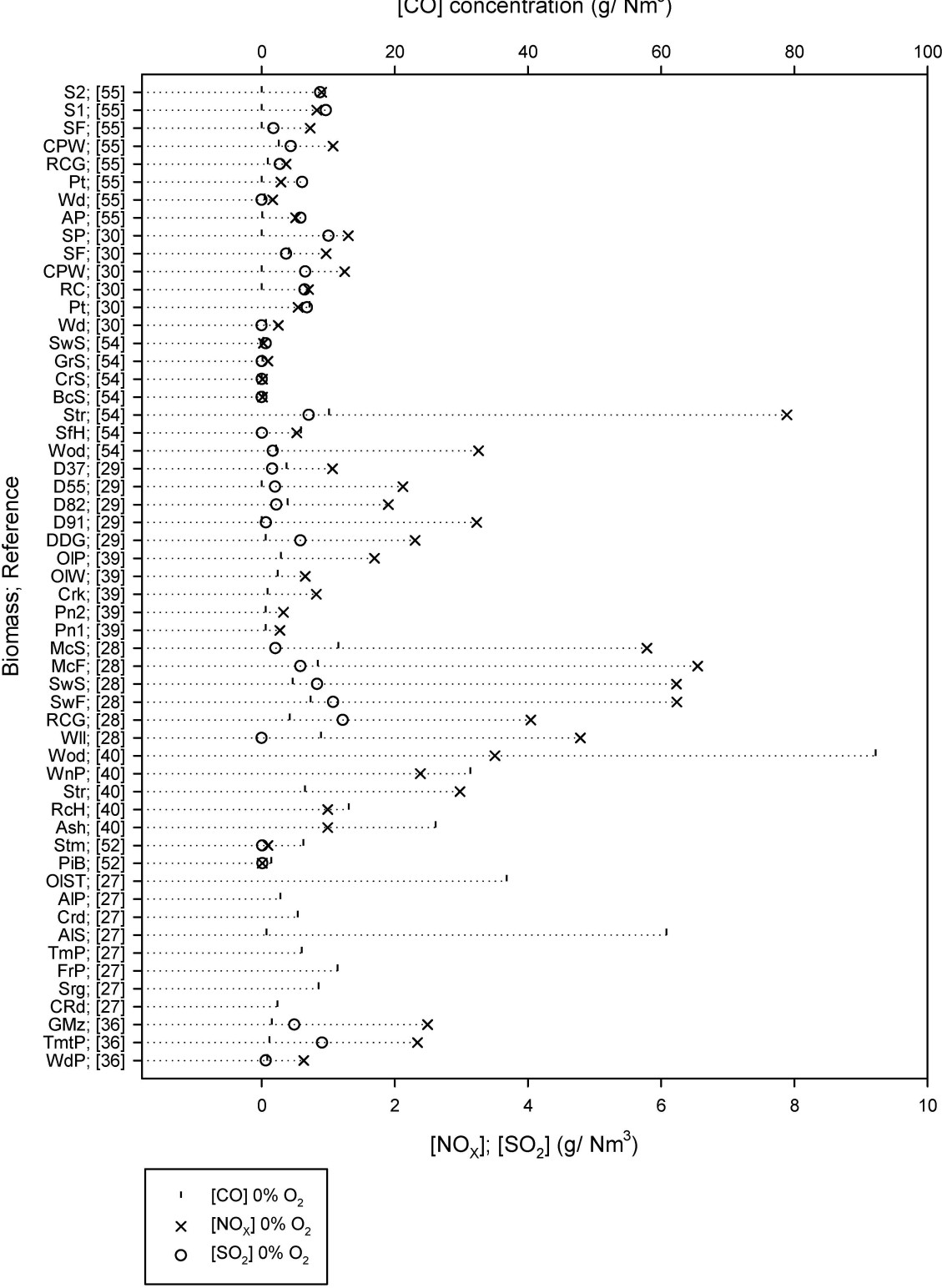

**Figure A12.** Toxic emission levels (CO, $NO_X$, and $SO_2$), as reported in previous bibliographical works studying several biomass types and combustion device thermal outputs (see Tables A2 and A3).

**Table A2.** Abbreviations used on the *Y*-axis of Figure A12 and the biomass type studied in the various works reviewed.

| Reference | Biomass | Abbreviation |
|---|---|---|
| [36] | Wood pellets | WdP; [36] |
| | Tomato pomace | TmtP; [36] |
| | Grape maize | GMz; [36] |
| [27] | Common reed | CRd; [27] |
| | Sorghum Sorghum | Srg; [27] |
| | Forest pellets | FrP; [27] |
| | Tomato pomace | TmP; [27] |
| | Almond shells | AlS; [27] |
| | Cardoon | Crd; [27] |
| | Almond prunings | AlP; [27] |
| | Almond shell peel | AlS; [27] |
| | Olive stones | OlST; [27] |
| [52] | Pine bark | PiB; [52] |
| | Stem wood | Stm; [52] |
| [40] | Almond shells | Ash; [40] |
| | Rice husks | RcH; [40] |
| | Straw | Str; [40] |
| | Wine pomace | WnP; [40] |
| | Wood pellets | Wod; [40] |
| [28] | Willow | Wll; [28] |
| | Red canary grass | RCG; [28] |
| | Switchgrass (Fall) | SwF; [28] |
| | Switchgrass (Spring) | SwS; [28] |
| | Miscanthus (Fall) | McF; [28] |
| | Miscanthus (Spring) | McS; [28] |
| [39] | Pine (1) | Pn1; [39] |
| | Pine (2) | Pn2; [39] |
| | Cork | Crk; [39] |
| | Olive wood | OlW; [39] |
| | Olive prunings | OlP; [39] |
| [53] | Poplar woodchips | PpW; [53] |
| [29] | Dried distilled grain | DDG; [29] |
| | Dried distilled grain + Municipal waste solids (90–10%) | D91; [29] |
| | Dried distilled grain + Municipal waste solids (80–20%) | D82; [29] |
| | Dried distilled grain + Municipal waste solids (50–50%) | D55; [29] |
| | Dried distilled grain + Municipal waste solids (30–70%) | D37; [29] |

**Table A2.** *Cont.*

| Reference | Biomass | Abbreviation |
|---|---|---|
| [54] | Wood | Wod; [54] |
| | Sunflower stalks | SfH; [54] |
| | Straw | Str; [54] |
| | Buckwheat shells | BcS; [54] |
| | Cornstalk | CrS; [54] |
| | Grain screenings | GrS; [54] |
| | Sewage sludge | SwS; [54] |
| [30] | Wood | Wd; [30] |
| | Peat | Pt; [30] |
| | Reed canary grass | RC; [30] |
| | Citrus pectin waste | CPW; [30] |
| | Sunflower husk | SF; [30] |
| | Straw pellets | SP; [30] |
| [55] | Apple pomace waste Wood | AP; [55] Wd; [55] |
| | Peat | Pt; [55] |
| | Reed canary grass | RCG; [55] |
| | Citrus pectin waste | CPW; [55] |
| | Sunflower husks | SF; [55] |
| | Straw pellets 1 | S1; [55] |
| | Straw pellets 2 | S2; [55] |

**Table A3.** CO, NO$_X$, and SO$_2$ levels, and the thermal efficiency reported in the literature for biomass combustion.

| Reference | Equipment | Thermal Output (KW) | Biomass | [CO] 0% O$_2$ (mg/Nm$^3$) | [NO$_X$] 0% O$_2$ (mg/Nm$^3$) | [SO$_2$] 0% O$_2$ (mg/Nm$^3$) | Efficiency (%) |
|---|---|---|---|---|---|---|---|
| [36] | Pellet boiler | 12.0 | Wood pellets | 829.3 | 633.9 | 62.9 | |
| | | | Tomato pomace | 1186.4 | 2338.9 | 908.0 | |
| | | | Grape maize | 1512.4 | 2491.9 | 491.3 | |
| [27] | Pellet boiler | 12.0 | Common reed | 2358.9 | | | 84.0 |
| | | | Sorghum | 8562.3 | | | 85.3 |
| | | | Forest pellets | 11,398.9 | | | 90.5 |
| | | | Tomato pomace | 6006.3 | | | 91.0 |
| | | | Almond shells | 712.5 | | | 85.0 |
| | | | Cardoon | 5404.0 | | | 91.6 |
| | | | Almond prunings | 2804.9 | | | 88.3 |
| | | | Almond shell peel | 60,755.8 | | | 78.5 |
| | | | Olive stones | 36,798.0 | | | 89.7 |
| [52] | Small combustion device | 50.0 | Pine bark | 1476.9 | 8.6 | 12.3 | |
| | | | Stem wood | 6252.4 | 93.1 | 9.6 | |

**Table A3.** *Cont.*

| Reference | Equipment | Thermal Output (KW) | Biomass | [CO] 0% $O_2$ (mg/Nm$^3$) | [NO$_X$] 0% $O_2$ (mg/Nm$^3$) | [SO$_2$] 0% $O_2$ (mg/Nm$^3$) | Efficiency (%) |
|---|---|---|---|---|---|---|---|
| [40] | Tubular furnace (Lab. Scale) | | Almond shells | 26,125.0 | 992.8 | | |
| | | | Rice husks | 13,062.5 | 992.8 | | |
| | | | Straw | 6531.3 | 2978.3 | | |
| | | | Wine pomace | 31,350.0 | 2382.6 | | |
| | | | Wood pellets | 92,205.9 | 3503.8 | | |
| [28] | Biomass boiler | 29.0 | Willow | 8917.1 | 4786.4 | | |
| | | | Red canary grass | 4246.4 | 4042.1 | 1218.0 | |
| | | | Switchgrass (Fall) | 7336.3 | 6235.1 | 1073.6 | |
| | | | Switchgrass (Spring) | 4641.4 | 6228.3 | 832.0 | |
| | | | Miscanthus (Fall) | 8459.4 | 6549.1 | 583.3 | |
| | | | Miscanthus (Spring) | 11,496.8 | 5785.4 | 206.1 | |
| [39] | Pellet-fired boiler | 22.0 | Pine (1) | | | | |
| | | | Pine (2) | 599.2 | 273.2 | | |
| | | | Cork | 623.2 | 327.9 | | |
| | | | Olive wood | 886.8 | 819.7 | | |
| | | | Olive prunings | 2396.8 | 655.8 | | |
| [53] | Fired-bed boiler | 140.0 | Poplar woodchips | 2876.1 | 1694.1 | | 94.0 |
| [29] | Fluidized bed combustor; 6 Kg/h | | Dried distilled grain | | | | |
| | | | Dried distilled grain + Municipal waste solids (90–10%) | 592.5 | 2302.8 | 583.4 | |
| | | | Dried distilled grain + Municipal waste solids (80–20%) | | 3228.0 | 64.0 | |
| | | | Dried distilled grain + Municipal waste solids (50–50%) | 3891.0 | 1901.0 | 221.8 | |
| | | | Dried distilled grain + Municipal waste solids (30–70%) | | 2120.1 | 201.9 | |
| [54] | Pellet boiler | 35.0 | Wood | 3732.1 | 1063.7 | 159.5 | |
| | | | Sunflower stalks | 2178.8 | 3257.2 | 167.1 | |
| | | | Straw | 5925.4 | 526.6 | 3.6 | |
| | | | Buckwheat shells | 10,099.6 | 7886.3 | 707.9 | |
| | | | Corn stalk | 196.8 | 12.3 | 0.0 | |
| | | | Grain screenings | 16.5 | 10.6 | 1.9 | |
| | | | Sewage sludge | 141.0 | 93.8 | 1.8 | |
| [30] | Multi-fuel boiler, Reduced Load | 40.0 | Wood | 263.3 | 27.1 | 62.2 | 93.0 |
| | | | Peat | 36,376.6 | 527.8 | 143.3 | 94.1 |
| | | | Reed canary grass | | | | 91.7 |
| | | | Citrus pectin waste | 658.1 | 250.1 | | 89.9 |
| | | | Sunflower husks | | | | 89.1 |
| | | | Straw pellets | 7195.1 | 546.8 | 681.7 | 89.8 |
| | | | Apple pomace waste | | 704.2 | 644.7 | 91.0 |

<div align="center">**Table A3.** *Cont.*</div>

| Reference | Equipment | Thermal Output (KW) | Biomass | [CO] 0% $O_2$ (mg/Nm$^3$) | [NO$_X$] 0% $O_2$ (mg/Nm$^3$) | [SO$_2$] 0% $O_2$ (mg/Nm$^3$) | Efficiency (%) |
|---|---|---|---|---|---|---|---|
| [55] | Multi-fuel boiler, Reduced Load | 40.0 | Wood | | 1247.1 | 654.3 | |
| | | | Peat | 4114.2 | 967.3 | 368.7 | 90.0 |
| | | | Reed canary grass | | 1299.1 | 1002.9 | 89.8 |
| | | | Citrus pectin waste | 81.0 | 509.9 | 584.4 | 89.7 |
| | | | Sunflower husks | 424.2 | 166.8 | 0.0 | 89.0 |
| | | | Straw pellets 1 | | 289.4 | 609.4 | 88.9 |
| | | | Straw pellets 2 | 903.6 | 367.2 | 274.2 | 88.0 |

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
