# Peer review of "Boiler Combustion Optimization of Vegetal Crop Residues from Greenhouses"

_agronomy, doi:10.3390/agronomy11040626_

Round 1

Reviewer 1 Report

The authors did not fully appreciate the comments proposing to present the content of the boiler test report in the form of a conventional article and to adapt it to the theme of Agronomy journal. The title of this manuscript, most of the research and the findings are more relevant to energy-themed journals. To devote about 30 pages to the discussion of the results of a commercial boiler test fired with greenhouse plant waste is an overemphasis on the results and complicates the analysis of the content of this manuscript for audience. Moreover, the main conclusion is formulated in a way that is already well known and the aim of all tests is to adjust the boiler so that the air and fuel supply are properly balanced, ensuring the highest efficiency of the boiler and low emissions to the environment. This work does not offer any specific solutions to achieve this goal and can hardly be expected, as the tests are performed without standardized conditions for such purposes and even using unusual different definitions (eg. combustion efficiency, thermal efficiency, thermal and heating efficiency).

In my opinion, in order to maintain the scientific quality and reputation of this journal, the manuscript must be substantially reorganized, shortened, presented in the form of a conventional article and adapted to the theme of the journal Agronomy. Only then, it could be further explored.

Author Response

Authors Reply to the Review Report (Academic Editor 1)

Comment: The authors did not fully appreciate the comments proposing to present the content of the boiler test report in the form of a conventional article and to adapt it to the theme of Agronomy journal. The title of this manuscript, most of the research and the findings are more relevant to energy-themed journals. To devote about 30 pages to the discussion of the results of a commercial boiler test fired with greenhouse plant waste is an overemphasis on the results and complicates the analysis of the content of this manuscript for audience. Moreover, the main conclusion is formulated in a way that is already well known and the aim of all tests is to adjust the boiler so that the air and fuel supply are properly balanced, ensuring the highest efficiency of the boiler and low emissions to the environment. This work does not offer any specific solutions to achieve this goal and can hardly be expected, as the tests are performed without standardized conditions for such purposes and even using unusual different definitions (eg. combustion efficiency, thermal efficiency, thermal and heating efficiency). In my opinion, in order to maintain the scientific quality and reputation of this journal, the manuscript must be substantially reorganized, shortened, presented in the form of a conventional article and adapted to the theme of the journal Agronomy. Only then, it could be further explored.

Response: We respect the comments from the reviewer, but we don’t agree that the paper doesn’t follow the form of regular papers, or it is not suitable for this journal. The paper already contains all the sections included in regular papers on this and other journals. Regarding the aim of the paper, we consider it completely relevant for the Agronomy journal because it is devoted to demonstrating the reliability of reusing greenhouse crop residues as heat and CO2 sources in greenhouses. The number of papers in this respect is quite reduced, moreover when considering the utilization of real crop residues in pilot-scale boilers operating at real conditions. We agree with the reviewer that papers related to the use of biomass as an energy source can be more adequate for energy-related journals. Nevertheless, in this case, the inputs and outputs of the combustion process in addition to the scale and conditions are specific to greenhouses then we consider the paper appropriate for this journal.

Reviewer 2 Report

The work concerns the current topic, i.e. the use of biomass as a renewable energy source. 
The study uses various post-harvest residues from the greenhouse, which are burned in the furnace, 
and the composition of gases resulting from combustion is analyzed.
The authors really have a lot of knowledge about this topic especially about biomass boilers .
There are still many linguistic errors in the manuscript.
Take, for example, the sentence in line 44
L44: "It is aimedits  application in  heating inside greenhouses also."
the sentence could be replaced with, for example, such
It is also intended for heating inside the greenhouse.
L237: In some places there is "Error! Reference source not found."
Please correct that.

The work is still hard to read, in my opinion. Its structure needs to be improved.
Authors should make every effort to ensure that the manuscript is one whole with an ordered structure. 
I have the impression that the Authors are jumping from one topic to another. 
There are many sentences that require a more detailed explanation, e.g.
L56: "At the other hand, other biomass cases utilization is more extended, 
but it is required transporting them from other places (which usually are relatively distant)." I'm not sure what the Authors mean in this context.
L63: "For this reason, the generation of other compounds associated like SO2 and NOX 
result particularly interesting because their presence have relatively important negative impact on plants growing" 
Is it really negative? What is the result of these gases? 
Dion et al. (2011) says "extracting SO2 and NO emissions from flue gases to form ammonium sulphate as a by-product valuable to fertilizer markets"
L65: "At the same time, CO levels can be used as indicator of O2supply. TheC/ O 65rate is also related with the production of Organic Compounds."
Please explain or at least provide a reference.

Please improve the structure of Introduction. 
I can find elements of the background of the problem studied and objective of the investigation, and also
significance of your work also in the light of the big picture. 
I think that the research hypotheses and the tools used to verify them can be presented more clearly.

Figure 1. Please explain the abbreviation CI.
In Materials and Methods the paragraph numbering from 2.3 is quite unclear. 
The statistical analysis is repeated in sections 2.3 and 2.7. 
It applies to other dependent variables, but it is the same. 
CO, NOX and SO2 gases are also broken down into two points. In my opinion, again the structure can be imporoved.

Please also check the structure of supplementary materials and references in the requirements for authors.

Author Response

Authors Reply to the Review Report (Academic Editor 2)

The authors of this manuscript thank the reviewers for their comments/ suggestions. Thanks to them, several points have been identified for improvement. According to these comments/ suggestions, various modifications have been implemented. We think that these modifications should satisfy these comments/ suggestions.

Comment: The work concerns the current topic, i.e. the use of biomass as a renewable energy source. The study uses various post-harvest residues from the greenhouse, which are burned in the furnace, and the composition of gases resulting from combustion is analyzed. The authors really have a lot of knowledge about this topic especially about biomass boilers. There are still many linguistic errors in the manuscript. Take, for example, the sentence in line 44

L44: "It is aim edits application in heating inside greenhouses also." the sentence could be replaced with, for example, such It is also intended for heating inside the greenhouse.

Response: Agree with the reviewer, this sentence has been revised.

L237: In some places there is "Error! Reference source not found." Please correct that.

Response: Agree with the reviewer, these sentences have been revised.

The work is still hard to read, in my opinion. Its structure needs to be improved. Authors should make every effort to ensure that the manuscript is one whole with an ordered structure. I have the impression that the Authors are jumping from one topic to another. There are many sentences that require a more detailed explanation, e.g. L56: "At the other hand, other biomass cases utilization is ore extended, but it is required transporting them from other places (which usually are relatively distant)." I'm not sure what the Authors mean in this context.

Response: Agree with the reviewer the entire document has been revised. This specific sentence has also been modified.

L63: "For this reason, the generation of other compounds associated like SO2 and NOX result particularly interesting because their presence have relatively important negative impact on plants growing" Is it really negative? What is the result of these gases? In lines 63- 65 it is explained that the main effects are related to growing. Some additional details have been given. Dion et al. (2011) says "extracting SO2 and NO emissions from flue gases to form ammonium sulphate as a by-product valuable to fertilizer markets"

Response: Sulphur and nitrogen oxides can damage the plants reducing their performance if ambient concentrations overpass critical values. However, these critical values are not usually achieved. If damaging values are achieved, it would be needed to install additional abatement processes for these compounds. The first step for decreasing any contaminant generation is to decrease this last as much as possible. Thus, this combustion optimization should be the first step. Furthermore, the process scale for the treatment of some contaminant depends on the amount generated. For this reason, this optimization is also important. Besides, the addition of a posterior treatment would increase the cost of equipment. It should be considered that this alternative aims for new utilization for this biomass source. The cost of the equipment required is going to have an impact on the profitability of this process. This article aims to give an alternative for adding value to a source which now is a residue.

L65: "At the same time, CO levels can be used as indicator of O2supply. The C/ O 65 rate is also related with the production of Organic Compounds." Please explain or at least provide a reference.

Response: Agree with the reviewer, two reference has been provided about this affirmation.

Please improve the structure of Introduction. I can find elements of the background of the problem studied and objective of the investigation, and also significance of your work also in the light of the big picture. I think that the research hypotheses and the tools used to verify them can be presented more clearly.

Response: Agree with the reviewer, the hypothesis has been highlighted in the introduction.

Figure 1. Please explain the abbreviation CI. In Materials and Methods, the paragraph numbering from 2.3 is quite unclear.

Response: Agree with the reviewer, this sentence has been revised.

The statistical analysis is repeated in sections 2.3 and 2.7. 

Response: Agree with the reviewer, these sections have been revised.

It applies to other dependent variables, but it is the same.  CO, NOX and SO2 gases are also broken down into two points. In my opinion, again the structure can be improved.

Response: Agree with the reviewer the sections commented had been aggregated, but we have preferred to let the rest of the variables organized in various sections.

Please also check the structure of supplementary materials and references in the requirements for authors.

Response: We understand this comment. We think that it is better to give some brief explanation about the results in the main article and further explanation in the supplementary material for those readers who would be interested. Nevertheless, some contents have been reorganized. The CO, NOX and SO2 gasses section in the supplementary material has been merged to have similar structure with respect to the main paper. Moreover, the supplementary material has been reorganized.

Round 2

Reviewer 1 Report

OK

Reviewer 2 Report

The authors improved the work.
Due to the fact that the work is valuable
and contains many important technical details,
I recommend it for publication.

This manuscript is a resubmission of an earlier submission. The following is a list of the peer review reports and author responses from that submission.

Round 1

Reviewer 1 Report

The presented article does not fully comply with the format of the conventional article and recalls the technical report, which presents the test results of non-standard applications of a specific purpose water heating boiler using part of the emitted CO2 gas in combustion products to stimulate plant vegetation. The introduction describes the research goals and objectives are vaguely worded. It is difficult to understand from the article whether the authors challenge is the efficient incineration of vegetable waste from greenhouse after harvest, while using part of the CO2 in combustion products to stimulate plant growth or make proposals to modify the boiler design for efficient incineration of tomato waste. The chosen research method differs significantly from the standard methods in testing the operating parameters of water heating boilers by changing the fuel, which will make it difficult to compare the results obtained. The tests were performed by burning the boiler with suitable fuel for wood pellets and olive stones, as well as unsuitable fuel (tomato waste and their mixtures with wood fuel) and changing the parameters of fuel and primary / secondary air supply. As expected, this agro-waste was found to burn, as summarized in the conclusion (From the data obtained, it can be argued that the O2-to-fuel rate has a significant impact on combustion efficiency and the emissions generated, 279-280), which confirms the known fact that the efficiency and pollution of the boiler is determined by the proper supply of air to the combustion chamber. The conclusions provide only abstract summaries and not concrete results and suggestions that could have scientific and practical value.

In accordance with established practice, boilers are tested in accordance with the procedures laid down in the standards, taking into account their scope, design, type of fuel used and requirements for their efficiency and emission levels. These standards describe the test conditions, durations and accuracies, which allow the results obtained to be repeated, if necessary, and compared with studies of similar boiler characteristics by other researchers. During the testing of boilers, it is very important to ensure the stability of the boiler operation for a certain set period of time in order to correctly assess the efficiency and measure the emissions. The article does not provide a hydraulic / thermal diagram of the boiler connection, it does not provide information about the time during which the boiler was operating in a constant mode under different loads. Typically, when testing boilers, in addition to the pollution parameters given in the article, organic gaseous compounds (OGC) and particular matter (PM) are usually measured. The choice of emissions is not explained. The present article does not clearly specify which part or parts of the tomato fruit bush were used for fuel production, as different parts may have different properties. There is a lack of information about what fuel was made from this biomass, whether it was pellets. If it was pellets, then what diameter and medium length the pellets were. There is a lack of information on the properties of this fuel, especially ash melting. This is an important characteristic to determine whether the right fuel can be produced from this pure biomass.

Structurally, the article has binding sections that take up 11 pages and appendices as many as 18 pages. The richness of the information provided and the frequent references to the annexes make the analysis of the article very difficult.

Reviewer 2 Report

As I understand it, the aim of the article is to optimize the boiler configuration for combustion pine pellets, olive pits and especially a mixture of tomato leftovers from the greenhouse.

The article has several disadvantages. Above all, English needs a lot of improvement. It's hard to read and grasp its sense.

I'm not sure if it is suitable for agronomy? Perhaps the authors might consider a technology journal.

How important is greenhouse biomass combustion compared to other biomass sources?

Figure 1 needs a description. What are the numbers? Other figures also need more explanation. The numbers on the X axis are only explained in the Results & Discussion section.

The Results & Discussion section should be separate and clearly highlighted.

It would be good if the article was sufficient to understand the research problem. Additional materials should complement the interested reader. It should not be the case that reading additional material is required to understand the article.

Unfortunately, I cannot recommend this article for consideration in its present form.